# Nucleophagy is promoted by two autophagy receptors and inhibited by chromatin-nuclear envelope tethering in fission yeast

Zhu-Hui Ma [1], Zhao-Qian Pan[1], Zhao-Di Jiang [1,2], Guang-Can Shao[1], Yu Hua [1], Fang Suo[1], Chen-Xi Zou[1], Yi-Feng Jiang[3], Meng-Qiu Dong [1,2] & Li-Lin Du [1,2] ✉

Selective autophagy of the nucleus, known as nucleophagy, targets nuclear components for degradation. The molecular mechanisms underlying nucleophagy remain inadequately understood. In this study, we identify a nucleophagy receptor, Npr1, in the fission yeast *Schizosaccharomyces pombe*. Npr1 is an Atg8-binding multi-transmembrane protein localized to the outer nuclear membrane. It functions redundantly with another autophagy receptor, Epr1, to promote nitrogen starvation-induced nucleophagy. In the absence of both Npr1 and Epr1, starved cells exhibit abnormal nuclear morphology and reduced survival. During nucleophagy, the nuclear envelope (NE) forms outward protrusions where Atg8 co-localizes with Npr1 and/or Epr1. These protrusions subsequently detach from the NE, resulting in the formation of autophagosomes that contain nucleophagy cargo. Notably, artificially enhancing chromatin association with the inner nuclear membrane leads to NE protrusions that fail to detach, thereby aborting nucleophagy. Our findings provide mechanistic insights into nucleophagy and suggest that abortive nucleophagy protects chromatin from degradation.

Macroautophagy (hereafter referred to as autophagy) plays a crucial role in maintaining intracellular homeostasis by transporting cytoplasmic components to lysosomes or vacuoles for degradation[1–3]. During autophagy, various cargos, including organelles, are enclosed by an expanding membrane called the phagophore (also known as the isolation membrane), which eventually forms a double-membrane-bound vesicle known as the autophagosome[4,5]. The fusion of the autophagosome with the lysosome or vacuole leads to the degradation of its internal contents[2,6]. Autophagy can be classified into non-selective and selective types[2,3]. Selective autophagy achieves cargo selectivity by employing proteins known as autophagy receptors[7]. Autophagy receptors share a common feature: their ability to interact with the Atg8/LC3 family of proteins through a sequence termed the Atg8-interacting motif (AIM) or the LC3-interacting region (LIR). The core sequence of the AIM/LIR is W/F/YxxL/V/I[8–10]. By simultaneously associating with their cognate cargos and Atg8/LC3 proteins, autophagy receptors establish a physical link between the cargos and the phagophore, thereby promoting selective cargo encapsulation into the autophagosome.

Selective autophagy targets various organelles for degradation, including mitochondria (mitophagy), the endoplasmic reticulum (ER-phagy), and the nucleus (nucleophagy)[11]. The first nucleophagy receptor, Atg39, was discovered in *Saccharomyces cerevisiae* and is found exclusively in budding yeasts[12]. Atg39 is a transmembrane protein localized at the nuclear envelope (NE) and interacts with Atg8 through an AIM in its N-terminal cytosolic tail[12]. In cells undergoing nucleophagy, Atg39 forms bright Atg8-positive puncta in a manner dependent on its Atg8-binding ability[13]. While Atg39 is essential for the selective autophagy of proteins in the outer nuclear membrane (ONM), inner nuclear membrane (INM), nucleolus, and nucleoplasm[12,13], it is dispensable for the autophagic degradation of nuclear pore components[14–16]. Instead, recent studies have identified the

[1]National Institute of Biological Sciences, Beijing, China. [2]Tsinghua Institute of Multidisciplinary Biomedical Research, Tsinghua University, Beijing, China. [3]ZEISS Microscopy Customer Center, Beijing Laboratory, Beijing, China. ✉e-mail: dulilin@nibs.ac.cn

nucleoporin Nup159 as an Atg8-binding protein that promotes the autophagic degradation of nuclear pore components in *S. cerevisiae*[14,15]. The selectivity factors participating in the autophagic degradation of nuclear components in other organisms remain largely undefined.

During nucleophagy in *S. cerevisiae*, histones and DNA are notably absent from nucleophagy-derived autophagosomes[13]. Proteomic analyses also revealed a significant depletion of histones in the contents of autophagic bodies in *S. cerevisiae*[17]. These findings suggest that nucleophagy in *S. cerevisiae* selectively avoids chromatin. However, it remains unclear how this selective exclusion of chromatin is achieved.

In the fission yeast *Schizosaccharomyces pombe*, the soluble autophagy receptor Epr1, which is localized to the ER (including the NE) through its interaction with the integral ER membrane proteins Scs2 and Scs22, mediates ER stress-induced ER-phagy and nucleophagy[18,19]. However, Epr1 is dispensable for nitrogen starvation-induced ER-phagy and nucleophagy[19]. In this study, we identify Npr1, a multi-transmembrane protein localized to the ONM, as a nucleophagy receptor that functions redundantly with Epr1 to facilitate the degradation of nuclear components during nitrogen starvation. Our study also reveals that, like the situation in *S. cerevisiae*, the chromatin is excluded from nucleophagy in *S. pombe*. Remarkably, we found that artificially tethering the chromatin to the INM inhibits nucleophagy at the step of NE fission, suggesting a mechanism that prevents chromatin degradation by inhibiting nucleophagy.

## Results

### Nitrogen starvation induces autophagy of specific nuclear components in *S. pombe*

To investigate nucleophagy in *S. pombe*, we examined whether nitrogen starvation triggers the autophagic degradation of specific nuclear components. For this purpose, we analyzed the autophagic processing of five fluorescent protein-tagged markers localized to distinct nuclear subcompartments: Pus1-mECitrine, a nucleoplasmic protein; mECitrine-Bqt4, an INM protein; mECitrine-Nup82, a nucleoporin; Ker1-mECitrine, a nucleolar protein; and H4-mECitrine, a histone protein associated with chromatin. In wild-type cells, but not in the autophagy-defective *atg5Δ* cells, substantial levels of Pus1-mECitrine (Fig. 1a), mECitrine-Bqt4 (Fig. 1b), mECitrine-Nup82 (Fig. 1c), and Ker1-mECitrine (Fig. 1d) were processed into free mECitrine after 24 h of nitrogen starvation. In contrast, the chromatin marker H4-mECitrine showed no evident processing in wild-type cells (Fig. 1e). These findings indicate that nitrogen starvation in *S. pombe* induces the autophagic degradation of the nucleoplasm, NE, nuclear pores, and nucleolus, but not chromatin.

### Identification of Npr1 as a candidate nucleophagy receptor

To identify autophagy receptors involved in nitrogen starvation-induced selective autophagy in *S. pombe*, we employed the TurboID-based proximity labeling technique to search for Atg8-binding proteins[20]. Mass spectrometry analysis of biotin-labeled proteins from nitrogen-starved cells expressing TurboID-tagged Atg8 revealed the enrichment of an uncharacterized protein SPCC70.04c (Fig. 2a and Supplementary Data 1), which we named Npr1 (for nucleophagy receptor). Co-immunoprecipitation and yeast two-hybrid (Y2H) analyses showed that Npr1 indeed interacts with Atg8 (Fig. 2b, c). Co-immunoprecipitation data revealed that Npr1 associates with Atg8 to a similar extent under both nitrogen-replete and nitrogen-starvation conditions (Fig. 2b), indicating that the Npr1–Atg8 interaction is not dependent on nitrogen starvation.

A proteome-wide study had previously shown that Npr1, when overexpressed, localizes to the NE[21]. Consistently, we found that endogenously fluorescent protein-tagged Npr1 exclusively localized to the NE in log-phase vegetative cells (Fig. 2d, e). Notably, its distribution on the NE was uneven, with a tendency to avoid overlap with nuclear

pores (Fig. 2e). The underlying reason for this uneven distribution remains unclear. Membrane protein prediction analysis indicated that Npr1 is an integral membrane protein containing four transmembrane helices, with both its N- and C-terminal tails facing the cytoplasm or nucleoplasm (Supplementary Fig. 2a)[22]. To experimentally assess the membrane topology, we established a split-GFP assay (Supplementary Fig. 2b)[23–25]. This assay utilizes two fragments of GFP, $GFP_{1-10}$ and $GFP_{11}$, which can assemble into a fluorescent protein if present in the same cellular compartment. By fusing $GFP_{1-10}$ to proteins of known localization (cytosolic Sum3 or ER luminal Gbs1) and fusing $GFP_{11}$ to either terminus of Npr1, we found that fluorescence was observed only when the Npr1 fusion constructs were co-expressed with the cytosolic reporter, indicating that both termini of Npr1 face the cytosol (Supplementary Fig. 2b). This orientation suggests that Npr1 localizes to the ONM, a notion supported by electron microscopy (EM) analysis using a genetically encoded EM tag, MTn[26], which showed that MTn-tagged Npr1 was distributed along the ONM (Supplementary Fig. 2c).

To further investigate the interaction between Npr1 and Atg8, we utilized AlphaFold2-Multimer, a tool that effectively predicts binding interfaces between Atg8 and its interacting proteins[27,28]. The predicted structure of the Atg8-Npr1 complex revealed that Npr1 employs a canonical AIM, specifically [22]WIDV[25], to bind Atg8 (Supplementary Fig. 2d). Mutations of key AIM residues (W22A/V25A) disrupted Npr1's interaction with Atg8, as demonstrated by both co-immunoprecipitation and Y2H analyses (Fig. 2b, c). These findings demonstrate that Npr1 interacts with Atg8 through an AIM located in its N-terminal tail facing the cytosol (Fig. 2f, g).

Under nitrogen starvation conditions, wild-type Npr1, but not its AIM-mutated variant, formed bright puncta that colocalized with Atg8 at the NE within 2 h (Fig. 2h). After 6 h of starvation, wild-type Npr1 showed pronounced vacuolar re-localization, while the AIM-mutated version exhibited significantly reduced vacuolar targeting (Fig. 2i, j).

Taken together, these results suggest that Npr1 is a candidate nucleophagy receptor. It localizes to the ONM, binds Atg8 via an AIM in its cytosolic N-terminal tail, and undergoes nitrogen starvation-induced vacuolar re-localization in an Atg8-binding-dependent manner.

### Npr1 and Epr1 are redundantly required for starvation-induced nucleophagy

To investigate the role of Npr1 in nucleophagy, we examined the effects of *npr1* deletion on the autophagic processing of various nuclear components during nitrogen starvation. Deletion of *npr1* did not substantially affect the processing of the nucleoplasmic protein Pus1-mECitrine, the INM protein mECitrine-Bqt4, the nucleolar protein Ker1-mECitrine, or the nucleoporin mECitrine-Nup82 (Fig. 3a and Supplementary Fig. 3a). However, overexpression of Npr1 from the *P41nmt1* promoter enhanced the autophagic processing of Pus1-mECitrine, particularly at the early time points of nitrogen starvation, suggesting that Npr1 has a nucleophagy-promoting function (Supplementary Fig. 3b).

We hypothesized that the absence of a nucleophagy defect in the *npr1Δ* mutant may be due to redundancy with another autophagy receptor. In *S. pombe*, the ER-phagy receptor Epr1 has been shown to be essential for ER-phagy and nucleophagy under ER stress but not during nitrogen starvation[18,19]. Thus, we reasoned that Npr1 and Epr1 may act redundantly under nitrogen starvation conditions. Indeed, deletion of both *npr1* and *epr1*, but not *epr1* alone, nearly completely abolished the nitrogen starvation-induced processing of Pus1-mECitrine, mECitrine-Bqt4, and Ker1-mECitrine (Fig. 3a and Supplementary Fig. 3a). Interestingly, the processing of the nucleoporin mECitrine-Nup82 remained unaffected in the *epr1Δ npr1Δ* mutant (Fig. 3a and Supplementary Fig. 3a). These results indicate that Npr1 and Epr1 redundantly function as nucleophagy receptors, promoting the autophagic degradation of nucleoplasmic, NE, and nucleolar

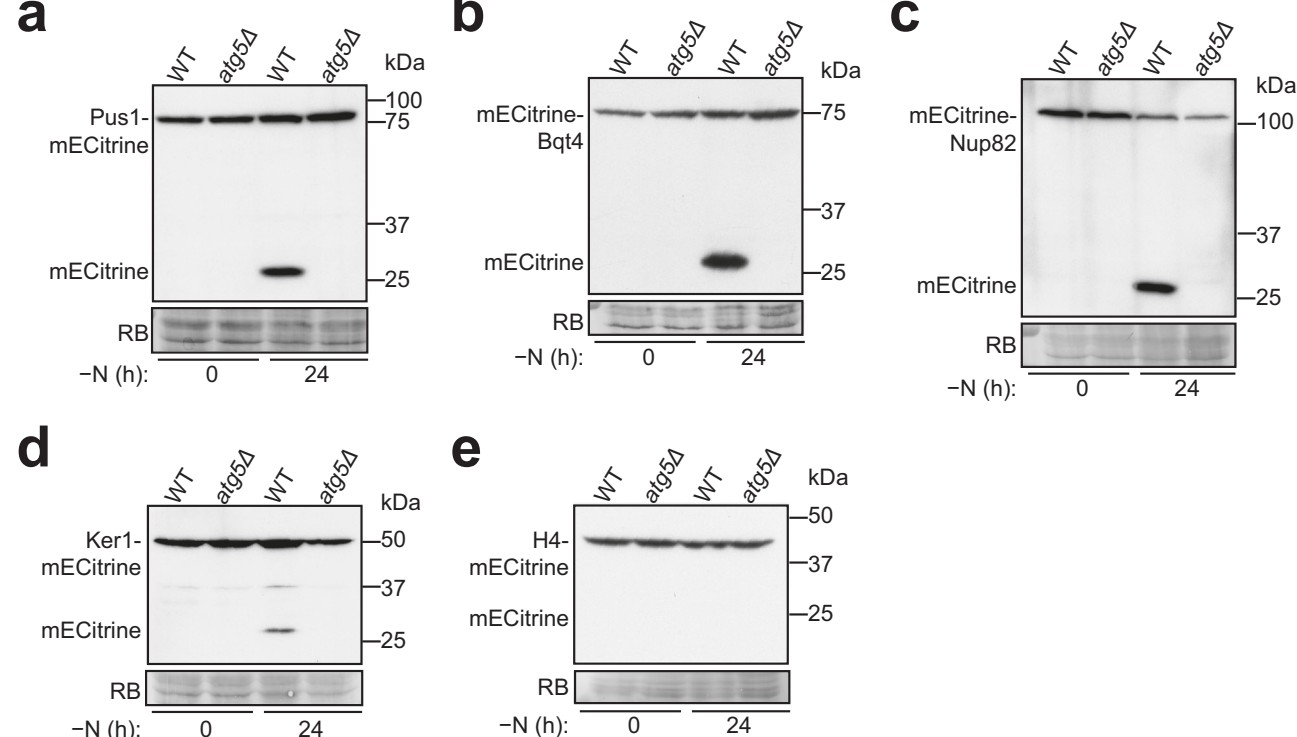

**Fig. 1 | Nitrogen starvation-induced autophagy targets nucleoplasmic, NE, nuclear pore, and nucleolar components, but excludes chromatin.** Nitrogen starvation-induced autophagic processing of the nucleoplasmic protein Pus1-mECitrine (**a**), INM protein mECitrine-Bqt4 (**b**), nucleoporin mECitrine-Nup82 (**c**), nucleolar protein Ker1-mECitrine (**d**), and histone H4-mECitrine (**e**) was examined in wild-type and *atg5Δ* cells. Cells expressing mECitrine-tagged proteins from the *P41nmt1* promoter were collected before and after 24 h of nitrogen starvation, and total lysates were analyzed by immunoblotting using an anti-GFP antibody that recognizes mECitrine. Post-immunoblotting staining of the PVDF membrane using Reactive Brown 10 (RB) served as the loading control. Experiments shown in this figure were independently repeated at least twice, consistently yielding similar results.

components, but not nuclear pores. This selectivity mirrors the situation in *S. cerevisiae*, where specialized autophagy receptors mediate nucleoporin degradation[14,15].

Consistent with their redundant roles, reintroducing either Npr1 or Epr1 into the *epr1Δ npr1Δ* mutant fully restored the autophagic processing of Pus1-mECitrine and mECitrine-Bqt4 (Fig. 3b and Supplementary Fig. 3c). In contrast, introducing AIM-mutated Npr1 (Npr1-W22A/V25A, abbreviated as Npr1*) or AIM-mutated Epr1 (Epr1-F352A/V355A, abbreviated as Epr1*) failed to rescue the nucleophagy defects (Fig. 3b and Supplementary Fig. 3c), demonstrating that the nucleophagy function of Npr1 and Epr1 depends on their abilities to bind Atg8. Using this reintroduction assay, we found that MTn-fused Npr1 and GFP11-fused Npr1 retain nucleophagy function (Supplementary Fig. 3d).

To further investigate the redundant relationship between Epr1 and Npr1, we tested whether the loss of one affects the expression level of the other. We found that the loss of Npr1 did not impact the protein level of Epr1, and conversely, the loss of Epr1 did not affect the protein level of Npr1 after 8 h of nitrogen starvation in the *isp6Δ psp3Δ* background, which blocks vacuolar degradation (Supplementary Fig. 3e).

Given that Epr1 localizes to both the cortical ER and NE[18], while Npr1 is exclusively localized to the NE, we hypothesized that only Epr1 contributes to cortical ER-phagy. Consistent with this idea, nitrogen starvation-induced autophagic processing of the cortical ER membrane protein Rtn1-mECitrine was moderately reduced in the *epr1Δ* mutant but was unaffected in the *npr1Δ* mutant (Supplementary Fig. 3f, g). Furthermore, the additional deletion of *npr1* did not exacerbate the mild phenotype observed in the *epr1Δ* mutant (Supplementary Fig. 3f,

g). These results suggest that other, yet unidentified, ER-phagy receptors may act redundantly with Epr1 during nitrogen starvation-induced cortical ER-phagy.

We previously showed that an artificial AIM (AIM^art) fused to the ER membrane protein Erg11 could functionally substitute for Epr1 in ER stress-induced ER-phagy[18]. To determine whether the nucleophagy functions of Npr1 and Epr1 could similarly be replaced, we employed the same AIM^art, composed of three tandem copies of the EEEWEEL sequence[29,30]. Substituting the N-terminal 29 amino acids of Npr1 (which contains its AIM) with AIM^art generated a chimeric protein, AIM^art-Npr1(30-244), which successfully rescued the nucleophagy defect of the *epr1Δ npr1Δ* mutant (Fig. 3c), indicating that AIM^art can functionally substitute for the AIM of Npr1.

We next fused AIM^art to various integral membrane proteins present on the NE. Only the ONM protein Kms1 and the ER membrane protein Erg11 with cytosol-facing AIM^art fusions rescued the nucleophagy defect of *epr1Δ npr1Δ* (Fig. 3c and Supplementary Fig. 3h). In contrast, fusion of AIM^art to the lumen-facing tails of these proteins, or to the nucleoplasm-facing N-terminus of the INM protein Man1 or the cytosol-facing C-terminus of the cortical ER membrane protein Rtn1, failed to rescue the defect. These findings indicate that the nucleophagy roles of Npr1 and Epr1 can be substituted by a membrane protein localized at the ONM, provided it possesses an AIM that faces the cytosol.

Taken together, these results demonstrate that Npr1 and Epr1 redundantly promote nitrogen starvation-induced nucleophagy by establishing a physical link between the ONM and Atg8. This link is mediated by their AIMs, which are essential for their function.

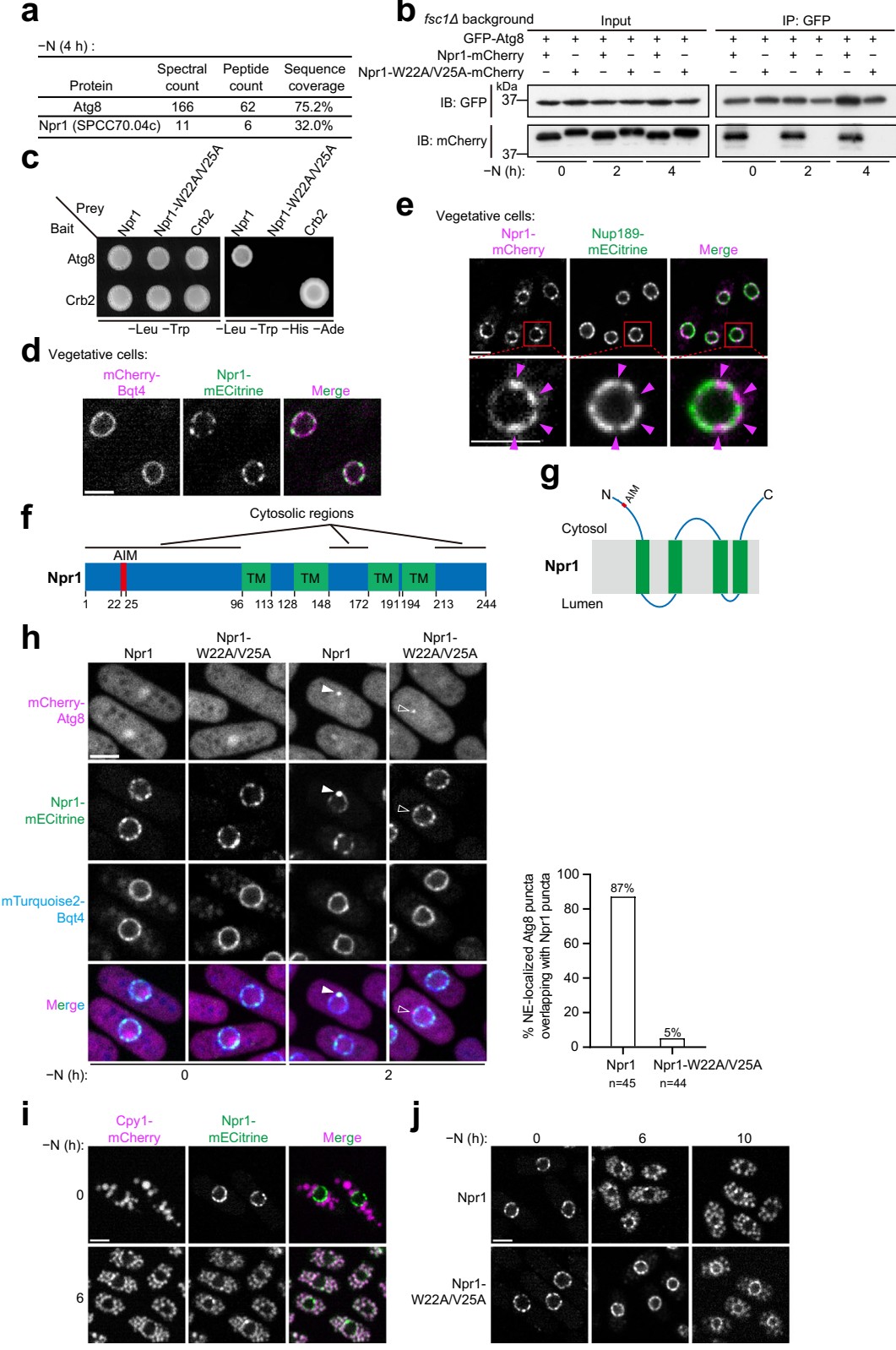

## Spatiotemporal analysis of Atg8 and nucleophagy receptors at the NE

To further investigate the mechanisms of nucleophagy, we used live-cell imaging to study the genetic interdependency, co-localization, and temporal dynamics of Atg8, Npr1, and Epr1. First, we observed that Atg8 puncta still formed at the NE in *epr1Δ npr1Δ* cells, albeit at slightly lower levels than in wild-type cells (Supplementary Fig. 4a, b).

Conversely, NE-localized Npr1 and Epr1 puncta were completely absent in *atg8Δ* cells, indicating that their formation strictly relies on Atg8 (Supplementary Fig. 4c, d).

Next, we analyzed the co-localization patterns of Atg8, Npr1, and Epr1 at the NE using time-lapse imaging of wild-type cells (Fig. 4). We classified NE-localized Atg8 puncta into four types based on their overlap with Npr1 and Epr1 for at least one time point: Type I (36%),

**Fig. 2 | Npr1 is an Atg8-binding NE protein that relocalizes to vacuoles during starvation in an Atg8-binding-dependent manner. a** Proximity labeling identified Npr1 (SPCC70.04c) as an Atg8 interactor. TurboID-tagged Atg8 (expressed under the *P41nmt1* promoter) was used to biotinylate proximal proteins in nitrogen-starved cells. Streptavidin-enriched proteins were analyzed using mass spectrometry. The full mass spectrometry results are presented in Supplementary Data 1. **b** Co-immunoprecipitation assays show that Atg8 binds wild-type Npr1 but not the AIM-mutated Npr1 (Npr1-W22A/V25A), both in nitrogen-replete conditions and after 2 or 4 h of nitrogen starvation. Assays were performed in an *fsc1Δ* strain background to prevent autophagic processing of GFP–Atg8 and Npr1 (Npr1-W22A/V25A)–mCherry. The experiment was independently repeated twice with similar results. **c** Y2H assays confirm that Npr1, but not Npr1-W22A/V25A, interacts with Atg8. **d** Npr1-mECitrine colocalizes with the NE marker mCherry-Bqt4. Bar, 3 µm. The experiment was independently repeated three times with similar results. **e** Npr1-mCherry shows a non-uniform NE distribution and tends not to overlap with the nucleopore marker Nup189-mECitrine. A single nucleus is shown in magnified

views below, with the positions of Npr1-mCherry signals highlighted by magenta arrowheads. Bar, 3 µm. The experiment was independently repeated three times with similar results. **f** Membrane topology of Npr1 was predicted using CCTOP. Detailed CCTOP output is shown in Supplementary Fig. 2a. Cytosolic regions were validated by split-GFP assays (Supplementary Fig. 2b). TM, transmembrane helices. **g** Schematic of Npr1's AIM motif and topology (not to scale). **h** AIM-dependent colocalization of Npr1 and Atg8 at the NE during starvation. Left: representative images show that wild-type Npr1, but not AIM-mutated Npr1, forms bright puncta overlapping with NE-localized Atg8 puncta (arrowheads). Bar, 3 µm. Right: quantification of the percentages of NE-localized Atg8 puncta overlapping with Npr1 puncta. The experiment was independently repeated twice with similar results. **i** Npr1 relocalizes to the vacuole during nitrogen starvation. Cpy1 serves as a vacuole lumen marker. Bar, 3 µm. The experiment was independently repeated twice with similar results. **j** AIM-mutated Npr1 exhibits a substantial delay in starvation-induced vacuole relocalization compared to wild-type Npr1. Bar, 3 µm. The experiment was independently repeated twice with similar results.

which overlapped with both; Type II (55%), with only Npr1; Type III (8%), with only Epr1; and Type IV (1%), with neither (Fig. 4a, e and Supplementary Fig. 4e, f). Thus, in wild-type cells, nearly all NE-localized Atg8 puncta overlapped with receptor puncta, suggesting that they represent nucleophagy events. The fact that over half of the NE-localized Atg8 puncta overlapped with only one receptor indicates that Npr1 and Epr1 can function independently, consistent with genetic data showing that either receptor is sufficient for nucleophagy.

We analyzed the sequential appearance and disappearance of Npr1, Epr1, and Atg8 at types I-III NE-localized Atg8 puncta (Fig. 4b–d). For type I puncta, Atg8 and Npr1 appeared simultaneously and before Epr1 in 50% of cases, while Atg8 appeared before both receptors in 34%. Regarding disappearance, Epr1 disappeared first in 47% of cases, while puncta of all three proteins disappeared simultaneously in 22%. Among type II puncta, Atg8 and Npr1 appeared simultaneously in 71% of cases, with Atg8 disappearing first in 67%. For type III puncta, Atg8 appeared first in 73% of cases, with Atg8 and Epr1 disappearing simultaneously in 73%. These results indicate that the sequential orders of the appearance and disappearance of Atg8, Npr1, and Epr1 puncta are not fixed.

All analyzed NE-localized Npr1 and Epr1 puncta (72 and 52, respectively) exhibited overlap with Atg8, consistent with the absence of these puncta in *atg8Δ* cells (Supplementary Fig. 4g). Overall, in 64% of cases, Atg8 appeared simultaneously with the receptors, while in 26%, Atg8 preceded receptor appearance. The rare cases (10%) where Npr1 preceded visible Atg8 puncta suggest that Npr1 puncta formation may require only low levels of Atg8. Notably, Npr1 puncta often persisted after Atg8 disappearance, indicating that their stability does not require continuous high levels of Atg8.

**Nucleophagy receptors promote NE protrusion formation**
During Atg39-mediated nucleophagy in *S. cerevisiae*, the NE forms outward protrusions[13,16,31]. To investigate whether NE protrusions also form during nucleophagy in *S. pombe*, we imaged nitrogen-starved cells expressing Npr1-mCherry and the INM protein mECitrine-Bqt4 (Fig. 4f). Time-lapse analysis revealed that in 52% of cases, Npr1 puncta formation was followed by a short outward projection of the mECitrine-Bqt4 signal at the same position, indicating NE protrusion formation (Fig. 4f, g). Notably, we did not observe any instances where NE protrusions formed before the emergence of Npr1 puncta at the same sites (Fig. 4h). Furthermore, the nucleoplasmic protein Pus1-mTurquoise2 co-localized with Bqt4 projections, indicating the presence of nucleoplasmic components within the NE protrusions (Supplementary Fig. 4h).

Similar observations were made with Epr1, where time-lapse analysis of cells expressing Epr1-ymScarlet2I and mECitrine-Bqt4 showed that in 54% of cases, the formation of NE-localized Epr1

puncta was associated with the subsequent formation of NE protrusions at the same sites (Supplementary Fig. 4i–k). These data suggest that the assembly of Npr1 and Epr1 puncta at the NE frequently precedes and potentially initiates the formation of NE protrusions.

Next, we investigated the relationship between Atg8 puncta and NE protrusions during nucleophagy. Time-lapse imaging of cells expressing ymScarlet2I-Atg8 and mECitrine-Bqt4 revealed that the formation of NE-localized Atg8 puncta was frequently followed by the formation of NE protrusions at the same locations (Supplementary Fig. 4n–p). EM analysis confirmed the presence of NE protrusions in nitrogen-starved cells (Fig. 4k and Supplementary Fig. 4s), revealing that these NE protrusions were closely surrounded by membranes that terminated at the necks of the protrusions—likely representing phagophores where lipidated Atg8 resides.

To assess the role of nucleophagy receptors in NE protrusion formation, we compared the frequency of Atg8-positive NE protrusions in wild-type and *epr1Δ npr1Δ* cells. We found that 40% of NE-localized Atg8 puncta were associated with NE protrusions in wild-type cells, compared to only 4% in *epr1Δ npr1Δ* cells (Supplementary Fig. 4o). These findings indicate that nucleophagy receptors play a critical role in promoting the formation of NE protrusions during nucleophagy.

**Distinct dynamics of nucleophagy components during NE protrusion release**
To investigate the fate of NE protrusions formed during nucleophagy, we analyzed time-lapse imaging data, focusing on sites where Atg8 and nucleophagy receptors assembled into puncta (Fig. 4f, i, j and Supplementary Fig. 4i, l–n, q, r). Using the INM protein mECitrine-Bqt4 as an NE marker, we observed that the disappearance of an NE protrusion was often accompanied by the emergence of a nearby Bqt4-labeled punctum in the cytosol (Fig. 4f). This suggests that NE protrusions are released into the cytosol.

Next, we investigated the dynamics of Npr1, Epr1, and Atg8 puncta in relation to NE protrusion release. For Npr1 puncta associated with NE protrusions in wild-type cells, 56% exhibited simultaneous release with the NE protrusion into the cytosol (Fig. 4i). Of the remaining 44%, 27% disappeared simultaneously with the NE protrusion, and 17% disappeared after the protrusion had vanished, without subsequent detection of cytosolic puncta. The released Npr1 puncta likely represent autophagosomes that have fully engulfed nuclear cargo and detached from the NE during nucleophagy. To assess whether the disappearance of Npr1 puncta represents entry into vacuoles, we examined the fate of Npr1 puncta and NE protrusions in the *fsc1Δ* background, where autophagosome-vacuole fusion is blocked[32]. We found that the fates of NE protrusion-associated Npr1 puncta in *fsc1Δ* cells were similar to those in wild-type cells, with 50% exhibiting

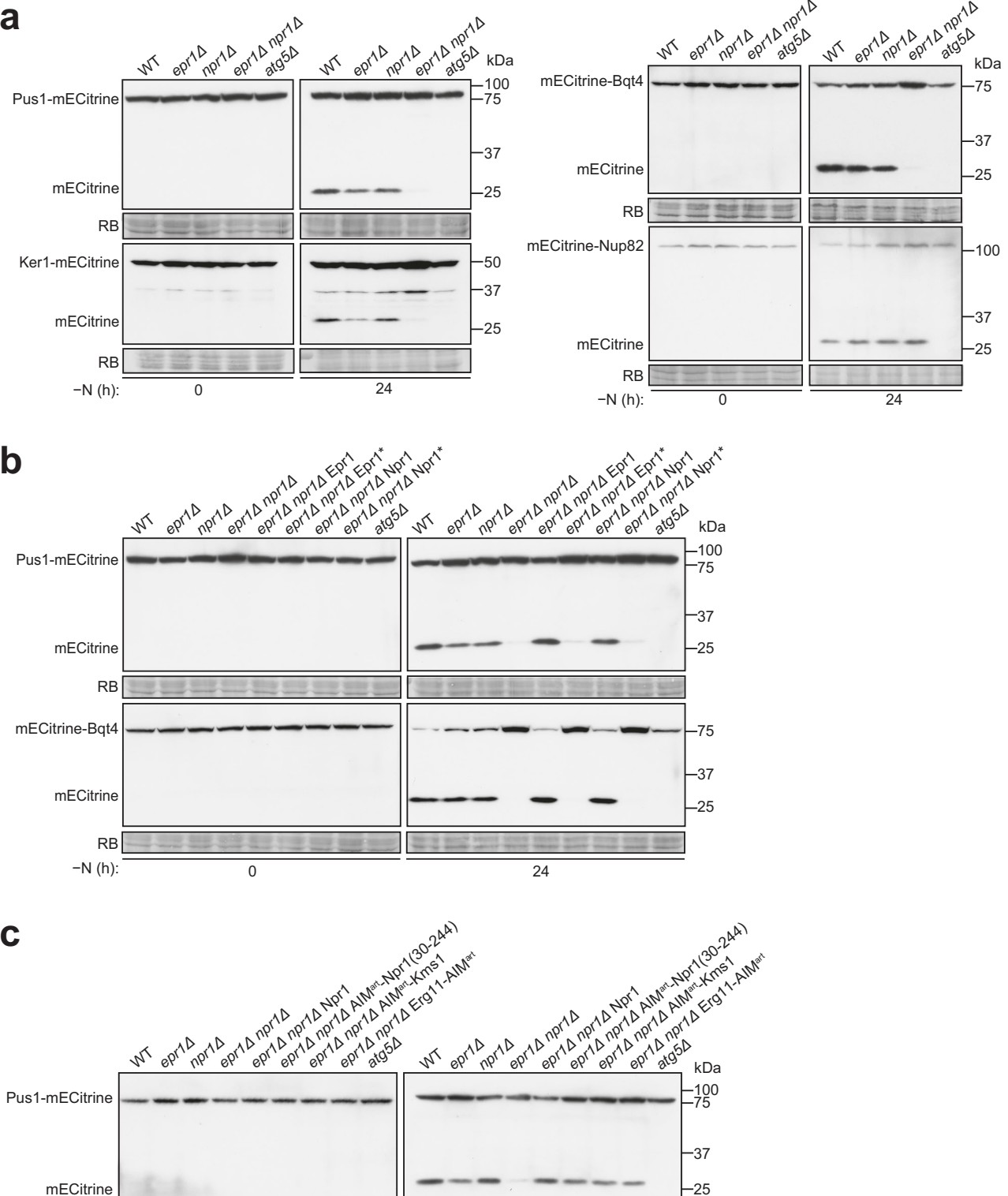

simultaneous release with the NE protrusion into the cytosol, 30% disappearing simultaneously with the NE protrusion, and 20% vanishing after the protrusion disappeared (Fig. 4i). These results suggest that the disappearance of Npr1 puncta likely represents premature termination of nucleophagy, rather than entry into vacuoles. Notably, Npr1 puncta not associated with NE protrusions were never observed to be released into the cytosol (Fig. 4j), suggesting that NE protrusion

formation is essential for successful nucleophagy-related autophagosome generation.

Epr1 puncta associated with NE protrusions exhibited three patterns: (1) simultaneous release with NE protrusions into the cytosol (30%); (2) disappearance before NE protrusion release (25%); and (3) disappearance of both from the NE without cytosolic puncta detection (45%, including 36% simultaneous disappearance and 9% NE

**Fig. 3 | Npr1 and Epr1 are redundantly important for starvation-induced nucleophagy. a** Nitrogen starvation-induced autophagic processing of the nucleoplasmic protein Pus1-mECitrine, the INM protein mECitrine-Bqt4, and the nucleolar protein Ker1-mECitrine, but not the nucleoporin mECitrine-Nup82, was abolished in *epr1Δ npr1Δ* cells. Post-immunoblotting staining of the PVDF membrane using RB served as the loading control. The blot image is representative of triplicate experiments. **b** The nucleophagy function of Npr1 and Epr1 is dependent on their AIMs. Ectopic expression of wild-type Epr1 or Npr1, but not AIM-mutated Epr1 (Epr1*) or Npr1 (Npr1*), in *epr1Δ npr1Δ* cells rescued the nucleophagy defect of *epr1Δ npr1Δ* cells. Epr1*, F352A/V355A; Npr1*, W22A/V25A. Epr1 and Epr1* were tagged with mCherry and expressed under the *P81nmt1* promoter. Npr1 and Npr1*

were tagged with mCherry and expressed under the *npr1* promoter. The experiment was independently repeated three times with similar results. **c** Fusing an artificial AIM (AIM^art), composed of three tandem copies of the EEEWEEL sequence, to either the N-terminally truncated Npr1 lacking its own AIM, or to the cytosol-facing N-terminus of Kms1, or to the cytosol-facing C-terminus of Erg11, rescued the nucleophagy defect of *epr1Δ npr1Δ* cells. AIM-fused proteins were tagged with mCherry and expressed under the *P41nmt1* promoter. Post-immunoblotting staining of the PVDF membrane using Coomassie Brilliant Blue (CBB) served as the loading control. The experiment was independently repeated three times with similar results.

protrusion disappearance first) (Supplementary Fig. 4l). Similar to Npr1, Epr1 puncta without associated NE protrusions were never released into the cytosol (Supplementary Fig. 4m).

The behavior of Atg8 puncta associated with NE protrusions differed notably—none were released simultaneously with NE protrusions into the cytosol. In 53% of instances, Atg8 puncta disappeared from the NE before the release of NE protrusions. In the remaining 47%, both disappeared from the NE, either simultaneously (20%) or with Atg8 puncta disappearing first (27%) (Supplementary Fig. 4q). Atg8 puncta not associated with NE protrusions were never observed to be released into the cytosol (Supplementary Fig. 4r).

Taken together, these findings reveal distinct behaviors among nucleophagy components during NE protrusion release. Atg8 typically disappears from the NE prior to protrusion release, while Npr1 tends to be released concurrently. Epr1 shows an intermediate pattern, being released together with NE protrusions in about half of the protrusion release events. Since NE protrusion release likely follows NE fission and autophagosome closure, these findings suggest that Atg8 is usually recycled before autophagosome closure, likely through Atg4-mediated delipidation[33]. It is important to note that the observed "disappearance" of Atg8 does not imply that all Atg8 molecules are recycled, but rather that the level of Atg8 associated with the closed autophagosome falls below the detection limit of microscopy. Conversely, Npr1, being an integral membrane protein, always remains within autophagosomes. As a peripheral membrane protein, Epr1 may escape from autophagosomes before their closure. Importantly, we observed two types of premature termination of nucleophagy in wild-type cells: the presence of nucleophagy receptor puncta without associated NE protrusions and the disappearance of NE protrusions without detectable cytosolic puncta, indicating that nucleophagy can abort naturally at either NE protrusion formation or release.

## Npr1- and Epr1-mediated nucleophagy maintains nuclear morphology and survival during nitrogen starvation

It is known that *S. cerevisiae* mutants defective in nucleophagy exhibit abnormal nuclear morphology during nitrogen starvation[12], and that *S. pombe* autophagy mutants display aberrant nuclear morphology during meiosis, which is induced by nitrogen starvation[34]. To investigate the role of nucleophagy in maintaining normal nuclear morphology during nitrogen starvation in *S. pombe*, we examined NE morphology using the INM protein mECitrine-Bqt4 as a marker.

Under nutrient-rich conditions, NE morphology appeared normal in both *atg5Δ* and *epr1Δ npr1Δ* mutants (Fig. 5a and Supplementary Fig. 5a, b). However, after 24 h of nitrogen starvation, these mutants exhibited striking NE aberrations characterized by projections with intensified mECitrine-Bqt4 signals. Other autophagy mutants, including *atg1Δ, atg13Δ, atg9Δ, atg14Δ, atg2Δ, atg18aΔ,* and *atg18bΔ*, also exhibited NE aberrations after nitrogen starvation (Supplementary Fig. 5a, b). The abnormal NE structures fell into two categories: extended NE projections, where the projections extend away from the NE, and ring-shaped NE projections, where both ends of a projection are associated with the NE. In *atg5Δ, atg13Δ, atg9Δ, atg14Δ, atg2Δ, atg18aΔ,* and *atg18bΔ* cells, extended NE projections were more

frequently observed than ring-shaped NE projections, whereas the *epr1Δ npr1Δ*, and *atg1Δ* mutants predominantly displayed ring-shaped projections. Neither type of aberration was observed in nitrogen-starved wild-type, *epr1Δ*, or *npr1Δ* cells (Fig. 5a and Supplementary Fig. 5a, b). This suggests that nucleophagy, redundantly promoted by Npr1 and Epr1, is required for maintaining normal nuclear morphology during nitrogen starvation. Supporting this, reintroducing either Npr1 or Epr1, but not their AIM-mutated forms, into the *epr1Δ npr1Δ* mutant restored normal NE morphology (Fig. 5a and Supplementary Fig. 5a, b). Reintroducing either artificial NE-Atg8 tethers, AIM^art-Kms1, or Erg11-AIM^art into the *epr1Δ npr1Δ* mutant also restored normal NE morphology (Supplementary Fig. 5a, b).

To further characterize these NE projections, we examined the localization of the nucleoplasmic protein Pus1-mECitrine after 24 h of nitrogen starvation (Fig. 5b). Pus1-mECitrine colocalized with mCherry-Bqt4 at the NE projections in both *atg5Δ* and *epr1Δ npr1Δ* cells, indicating that these projections are nucleoplasm-containing structures (Fig. 5b). The intensified Bqt4 signals observed at these projections likely result from the presence of two layers of NE sandwiching a thin layer of nucleoplasm.

To visualize the ultrastructures of the NE projections, we employed focused ion beam-scanning electron microscopy (FIB-SEM) (Fig. 5c). Wild-type cells exhibited circular-shaped nuclei in FIB-SEM slices and spherical-shaped nuclei in three-dimensional reconstructions (Fig. 5c and Supplementary Movie 1). In stark contrast, and consistent with our light microscopy observations, *epr1Δ npr1Δ* and *atg5Δ* cells exhibited aberrant nuclei with prominent projections in FIB-SEM slices, and these projections contain two closely spaced layers of NE sandwiching nucleoplasm (Fig. 5c). The space within the ring-like profile of the projections is cytoplasm, as evidenced by the presence of cytoplasmic membrane organelles, most prominently vacuoles (Fig. 5c, d). Since an *S. pombe* cell contains numerous small vacuoles dispersed throughout the cytoplasm, the presence of vacuoles within the ring-like profiles of NE projections may simply result from this; however, we cannot rule out the possibility of a preferential spatial relationship between NE projections and vacuoles. Additionally, three-dimensional reconstruction revealed that the ring-shaped NE projections observed under light microscopy and in FIB-SEM slices correspond to cross-sections of dome-like or semi-dome-like structures extending from the NE surface (Fig. 5c and Supplementary Movies 2 and 3). The interiors of these dome-like or semi-dome-like structures often contained a few vacuoles (Supplementary Movies 2 and 3).

To further investigate the changes in the NE in *epr1Δ npr1Δ* and *atg5Δ* cells compared to wild-type cells, we quantified the volume of the nucleus, as well as the surface area and volume of the NE from FIB-SEM slices. Our analysis revealed that the volume of the nucleus remained unchanged in *epr1Δ npr1Δ* and *atg5Δ* cells compared to wild-type cells. However, the surface area and volume of the aberrant NE in these mutants were significantly increased compared to wild-type cells (Supplementary Fig. 5c). These findings suggest that nucleophagy plays a crucial role in maintaining normal NE morphology during nitrogen starvation, and that nuclear deformation results in NE

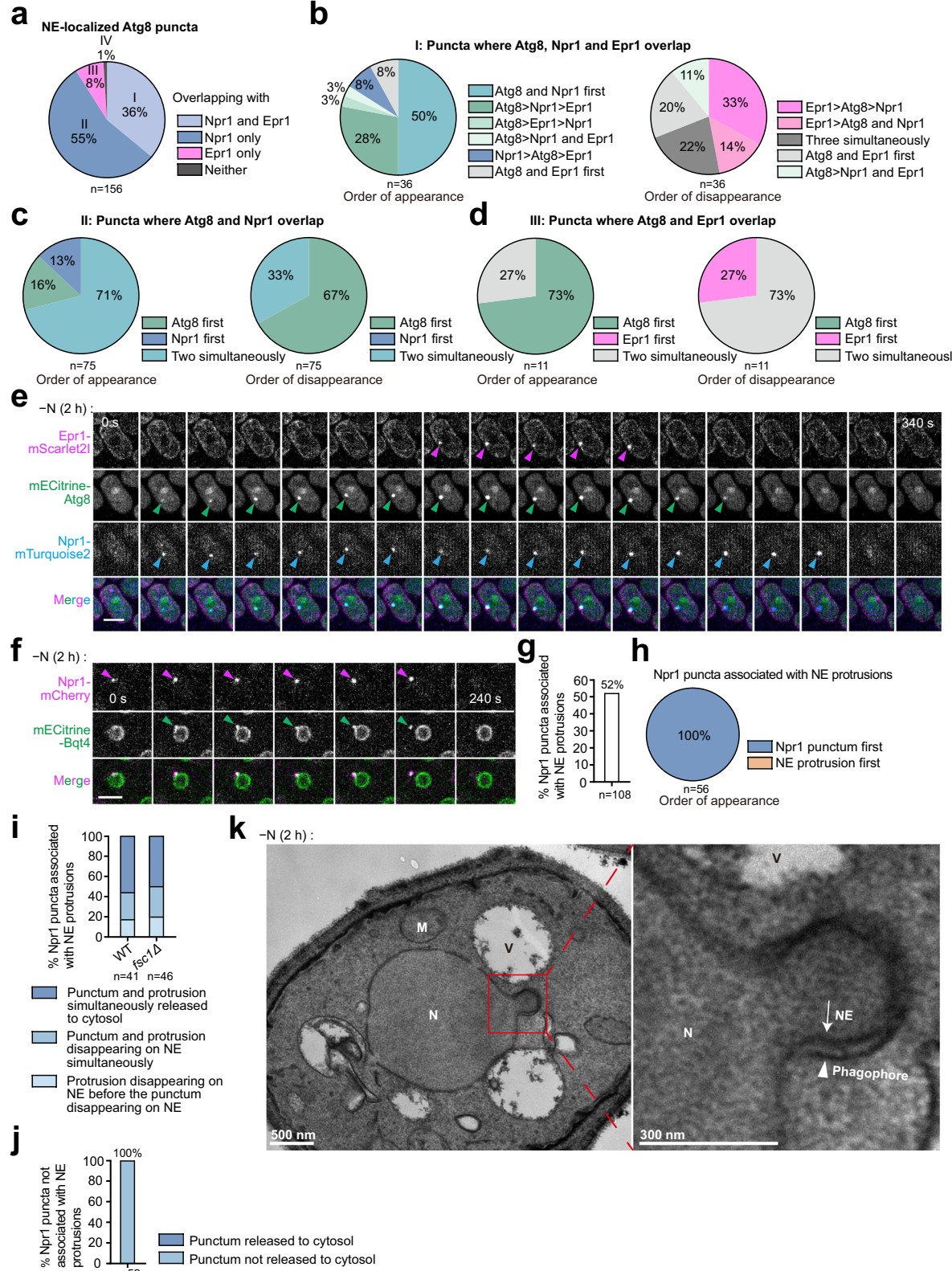

expansion without altering nuclear volume in nucleophagy-deficient cells.

Next, we investigated the contribution of nucleophagy to cell survival during nitrogen starvation using spot assays (Fig. 5e). After 3 days of nitrogen starvation, both *epr1Δ npr1Δ* and *atg5Δ* mutants showed reduced survival compared to wild-type cells, while *epr1Δ* or *npr1Δ* mutants were unaffected. This survival defect in *epr1Δ npr1Δ*

cells was rescued by reintroducing either Npr1 or Epr1, but not their AIM-mutated forms. Reintroduction of either artificial NE-Atg8 tethers, AIM^art-Kms1, or Erg11-AIM^art, into the *epr1Δ npr1Δ* mutant also rescued the survival defect (Supplementary Fig. 5d). Collectively, these results demonstrate that Npr1- and Epr1-mediated nucleophagy play essential roles in maintaining normal nuclear morphology and promoting cell survival during nitrogen starvation in *S. pombe*.

**Fig. 4 | Dynamics of nucleophagy-related Atg8, Epr1, and Npr1 puncta and NE protrusions. a** The overlap of NE-localized Atg8 puncta with Epr1 and Npr1 puncta in time-lapse imaging data. Time-lapse imaging was performed on cells co-expressing Epr1-mScarlet2I, mECitrine-Atg8, and Npr1-mTurquoise2 after 2 h of nitrogen starvation (20-s intervals). NE-localized Atg8 puncta, whose entire life-spans from appearance to disappearance were captured, were analyzed and categorized as: Type I (Epr1+/Npr1+), Type II (Npr1+ only), Type III (Epr1+ only), or Type IV (neither). **b–d** The order of appearance and disappearance of Npr1, Epr1, and Atg8 at types I (**b**), II (**c**), and III (**d**) NE-localized Atg8 puncta. **e** Representative time-lapse series of a cell containing a type I punctum. Epr1-mScarlet2I, mECitrine-Atg8, and Npr1-mTurquoise2 signals enriched at this punctum are denoted by arrow-heads. Bar, 3 μm. **f** NE protrusion release into the cytosol. A representative time-lapse series showing that a Bqt4-labeled NE protrusion associated with an Npr1

punctum was released into the cytosol. Bar, 3 μm. **g** Quantification of the percentage of Npr1 puncta associated with NE protrusions in wild-type cells after 2 h of nitrogen starvation. A total of 108 Npr1 puncta were analyzed by time-lapse imaging over their entire life spans, from appearance to disappearance. **h** The order of appearance of Npr1 puncta and associated NE protrusions. **i** Quantification of the percentages of Npr1 puncta and associated NE protrusions that were released into the cytosol or that disappeared from the NE without being detected in the cytosol in wild-type (WT) and *fsc1Δ* cells after 2 h of nitrogen starvation. **j** None of the Npr1 puncta that were not associated with NE protrusions were released into the cytosol. **k** A representative EM image of starved wild-type cells showing a phagophore (arrowhead) wrapping around an NE protrusion. N nucleus, V vacuole, M mito-chondrion. The experiment was independently repeated twice with similar results.

## Inhibition of nucleophagy by Lem2 overexpression

During our examination of various INM proteins as NE markers, we made a serendipitous discovery that the INM protein Lem2, when expressed exogenously from a medium-strength promoter (*P41nmt1*), nearly completely blocked the autophagic processing of Npr1-mECitrine and Pus1-mECitrine (Fig. 6a). Notably, this inhibition was specific to cargos of Npr1- and Epr1-mediated nucleophagy, as no effect was observed on the autophagic processing of the nucleoporin mECitrine-Nup82 (Supplementary Fig. 6a) and the cortical ER membrane protein Rtn1-mECitrine (Supplementary Fig. 6b). These findings suggest that elevated Lem2 expression selectively disrupts nucleophagy. We decided to investigate this phenomenon, as it implies the existence of a regulatory constraint on nucleophagy.

To determine the level of Lem2 required to inhibit nucleophagy, we expressed Lem2-mCherry using four different promoters with increasing expression strengths: *P81nmt1* < *Padf1* < *P41nmt1* < *Pcyc1* (Supplementary Fig. 6c, d). Interestingly, even a modest level of exogenous Lem2 expression (2–4-fold of the endogenous level) driven by the weakest promoter, *P81nmt1*, resulted in observable, albeit moderate, inhibition of nucleophagy. Expression from stronger promoters (*Padf1* and above), at levels 8–16-fold or higher than the endogenous level, resulted in near-complete inhibition of nucleophagy. These findings indicate that Lem2 inhibits nucleophagy in a dose-dependent manner.

To explore how this inhibition occurs, we examined the localization of the nucleophagy receptor Npr1 in Lem2-OE cells. After 8 h of nitrogen starvation, wild-type cells showed Npr1-mECitrine primarily in the vacuole lumen, while both *atg5Δ* and Lem2-OE cells retained Npr1-mECitrine at the NE (Fig. 6b). Notably, Lem2-OE cells displayed NE-localized Npr1-mECitrine puncta, unlike *atg5Δ* cells, where no such puncta were observed. This indicates that Lem2 overexpression permits the initial phase of nucleophagy—Npr1 puncta formation—but interferes with later stages.

Similar to wild-type cells, Npr1-mECitrine puncta in Lem2-OE cells were often associated with NE protrusions (Fig. 6b), which also contained the nucleoplasmic protein Pus1-mECitrine (Supplementary Fig. 6e). EM revealed that NE protrusions in Lem2-OE cells were surrounded by phagophores (Fig. 6c and Supplementary Fig. 6f), indicating they are nucleophagy intermediate structures.

Time-lapse imaging showed that in both wild-type and Lem2-OE cells, NE-localized Npr1 puncta were dynamic structures with lifetimes of several minutes (Fig. 6d and Supplementary Fig. 6h). These puncta were often associated with NE protrusions that formed after the puncta appeared (Fig. 6d, e and Supplementary Fig. 6g). However, while more than half of the NE protrusions in wild-type cells eventually detached from the NE and were released into the cytosol (Figs. 4i and 6f), NE protrusions in Lem2-OE cells never detached. Instead, they disappeared from the NE without being released into the cytosol (Fig. 6d, f). In most cases, NE protrusions disappeared simultaneously with Npr1 puncta, although in 23% of instances, NE protrusions

disappeared before Npr1 puncta (Fig. 6e). These findings indicate that Lem2 overexpression allows nucleophagy to progress to the stage of NE protrusion formation but blocks the subsequent step of protrusion release.

To further explore this phenomenon, we analyzed the behavior of Atg8 in Lem2-OE cells. While Atg8 and Npr1 co-localized at NE puncta, appearing simultaneously in most cases Supplementary Fig. 6h–j), Atg8 puncta consistently disappeared from the NE before Npr1 puncta (Supplementary Fig. 6j). Furthermore, Atg8 puncta always disappeared before the NE protrusions (Supplementary Fig. 6k), suggesting that components of the autophagy machinery dissociate before the retraction of NE protrusions during the abortive nucleophagy process.

## Chromatin-INM tethering inhibits nucleophagy

To investigate how Lem2 overexpression inhibits nucleophagy, we analyzed which regions of Lem2 are critical for this effect. Lem2 contains an N-terminal LEM domain (amino acids 1–60) that binds DNA[35], a Bqt4-binding motif (BBM, amino acids 261–279) that localizes Lem2 to the NE by interacting with the INM protein Bqt4[36], and two trans-membrane helices (Fig. 7a).

Using truncation analysis, we found that an N-terminal soluble fragment of Lem2 (amino acids 1-279), which includes the LEM domain and BBM, was sufficient to inhibit nucleophagy, whereas fragments lacking either domain were ineffective (Supplementary Fig. 7a). Internal deletion analysis confirmed that both domains are necessary for nucleophagy inhibition (Supplementary Fig. 7a). Furthermore, a fusion construct containing just the LEM domain and BBM inhibited nucleophagy, while neither domain alone was sufficient (Fig. 7b). These results establish that the combination of the LEM domain and BBM represents the minimal functional unit required for nucleophagy inhibition.

Because the LEM domain binds DNA and the BBM binds Bqt4, we hypothesized that their combination tethers chromatin to the INM, where Bqt4 resides. To test this hypothesis, we substituted the BBM in the fusion construct with the C-terminal transmembrane helix (TM, amino acids 412–432) of Bqt4, a single-pass INM protein with its N-terminus facing the nucleoplasm[37]. This LEM-Bqt4(TM) fusion localized to the NE (Supplementary Fig. 7d) and effectively inhibited nucleophagy (Fig. 7c, d), suggesting that the role of the BBM in nucleophagy inhibition is to mediate INM association.

To determine whether the LEM domain's role is chromatin association, we tested whether it could be replaced by other chromatin-binding domains. We substituted the LEM domain with either the histone-binding bromodomain (BD1) of *S. pombe* Bdf1 or the non-specific DNA-binding protein Sso7d from *Archaea* (Fig. 7c, d)[38,39]. Fusion proteins containing BD1 or Sso7d, along with the TM of Bqt4, localized to the NE (Supplementary Fig. 7d) and effectively inhibited nucleophagy (Fig. 7d). Notably, when the histone-binding ability of BD1 was disrupted by a point mutation (Y123F, BD1*)[40], or the DNA-binding ability of Sso7d was weakened by two mutations (W24A and

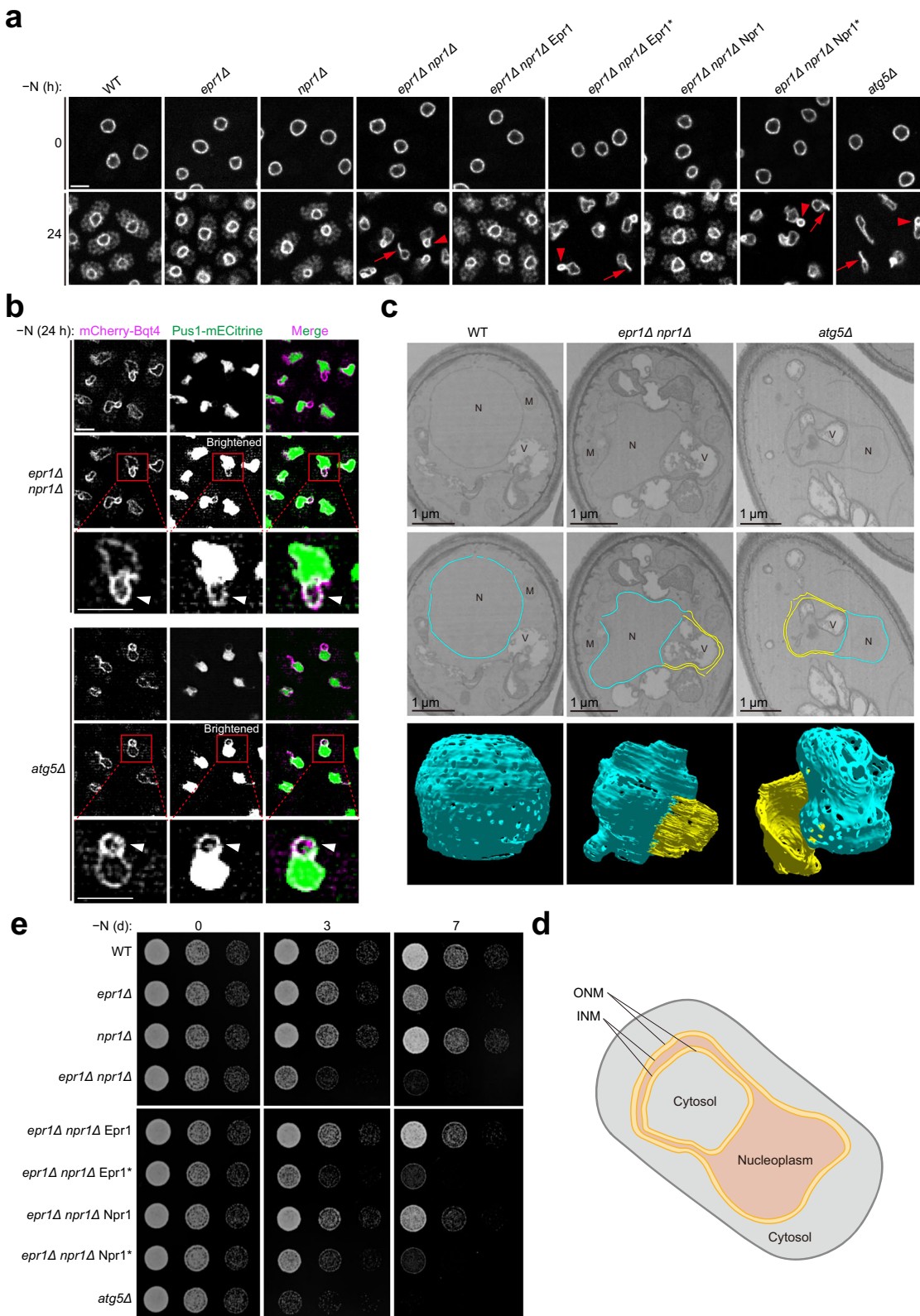

R43E, Sso7d*)[41], these fusion proteins no longer strongly inhibited nucleophagy (Fig. 7d). These results indicate that the main role of the LEM domain in nucleophagy inhibition is to mediate chromatin association.

To further solidify the role of chromatin-INM tethering, we tested Sso7d fused to the multi-transmembrane INM protein Bqt3, a binding partner of Bqt4[37]. This fusion also effectively inhibited nucleophagy

(Supplementary Fig. 7b, d). Notably, even though the known physiological functions of Bqt3 and Bqt4 require their interaction with each other[37,42], nucleophagy inhibition by Sso7d-Bqt3 occurred in the absence of *bqt4*, and nucleophagy inhibition by Sso7d-Bqt4(TM) occurred in the absence of *bqt3* (Supplementary Fig. 7b–d), suggesting that nucleophagy inhibition does not depend on Bqt3-Bqt4 interaction. Remarkably, the Sso7d-Bqt3 fusion protein was able to inhibit

**Fig. 5 | Npr1- and Epr1-mediated nucleophagy is essential for maintaining nuclear morphology and cell survival under nitrogen starvation. a** *epr1Δ npr1Δ* and *atg5Δ* cells formed NE projections during nitrogen starvation. This phenotype in *epr1Δ npr1Δ* was rescued by reintroducing either Epr1 or Npr1 in an AIM-dependent manner. Fluorescence microscopy was used to visualize cells expressing the INM protein mECitrine-Bqt4 before and after 24 h of nitrogen starvation. Representative ring-shaped NE projections are indicated by arrowheads, and extended NE projections by arrows. Bar, 3 μm. The experiment was independently repeated three times with similar results. **b** The nucleoplasmic protein Pus1-mECitrine colocalized with mCherry-Bqt4 at NE projections in *epr1Δ npr1Δ* and *atg5Δ* cells. Images were processed by deconvolution. Weak Pus1-mECitrine signals at NE projections became clearly visible only after brightness adjustments. In magnified views of the boxed areas, NE projections are indicated by arrowheads. Bar, 3 μm. The experiment was independently repeated three times with similar

results. **c** FIB-SEM analysis revealed the three-dimensional structures of NE projections in *epr1Δ npr1Δ* and *atg5Δ* cells. Top: representative FIB-SEM slices of a wild-type (WT) cell, an *epr1Δ npr1Δ* cell, and an *atg5Δ* cell. Middle: the same slices with the NE colored blue and NE projections colored yellow. N nucleus, M mitochondrion, V vacuole. Bottom: 3D reconstructions of the NE and NE projections in the cells shown in the top and middle rows. Approximately 300 FIB-SEM slices were used for each reconstruction. The NE is colored blue, and the NE projections are colored yellow. The experiment was independently repeated twice with similar results. **d** Schematic of NE projections in *epr1Δ npr1Δ* and *atg5Δ* cells after nitrogen starvation. **e** Reduced survival of *epr1Δ npr1Δ* and *atg5Δ* cells after nitrogen starvation. This phenotype of *epr1Δ npr1Δ* was alleviated by reintroducing either Epr1 or Npr1 in an AIM-dependent manner. Cells subjected to nitrogen starvation for 0, 3, and 7 days were plated in five-fold serial dilutions on YES plates, which were photographed after colony formation.

---

nucleophagy even when expressed from much weaker promoters, including the *P81nmt1* promoter (Supplementary Fig. 7e).

To rule out the possibility that nucleophagy inhibition depends on specific functions of Bqt4(TM) or Bqt3 beyond their INM localization, we employed an artificial INM protein h2NLS-LR2-WALP23, consisting of the NLS of budding yeast Heh2, a random linker sequence LR2, and an artificial transmembrane helix WALP23[43]. Fusion proteins combining BD1 or Sso7d with h2NLS-LR2-WALP23 inhibited nucleophagy, whereas fusion proteins with the mutated forms BD1* or Sso7d* did not, despite all constructs localizing to the NE (Supplementary Fig. 7f, g). This definitively demonstrates that simply tethering chromatin to the INM is sufficient to inhibit nucleophagy.

A key feature of nucleophagy inhibition by Lem2 overexpression is the blockage of NE protrusion release into the cytosol (Fig. 6h). We hypothesized that this blockage is caused by excessive chromatin tethering to the INM, leading to the presence of chromatin in the NE protrusions. Analysis of histone H3 localization revealed its presence in 7% of Npr1 puncta-associated NE protrusions in wild-type cells, increasing to 63% in cells OE Lem2 (Fig. 7e). Similarly, H3 was frequently detected in Npr1 puncta-associated NE protrusions in cells expressing Sso7d-h2NLS-LR2-WALP23 but not in cells expressing Sso7d*-h2NLS-LR2-WALP23 (Fig. 7e). Super-resolution microscopy confirmed H3 localization within Npr1-positive structures in Lem2-OE cells (Supplementary Fig. 7h). These findings suggest that chromatin presence in NE protrusions prevents their release, resulting in abortive nucleophagy.

We investigated whether nucleophagy inhibition by chromatin-INM tethering results in NE morphology aberrations and found that cells overexpressing Lem2 and those expressing Sso7d-h2NLS-LR2-WALP23, but not those expressing Sso7d*-h2NLS-LR2-WALP23, displayed abnormal NE morphology (Fig. 7f and Supplementary Fig. 8a). The most striking abnormal NE structures are ring-shaped NE projections, extended NE projections, and NE invaginations (Supplementary Fig. 8a, b). These findings indicate that alterations in NE morphology are a common consequence of all conditions that impede nucleophagy.

## Discussion

In this study, we identified Npr1, an Atg8-binding protein that localizes to the ONM and acts as a dedicated nucleophagy receptor. During nitrogen starvation, Npr1 functions redundantly with Epr1 to mediate the selective degradation of nuclear components. Both Npr1 and Epr1 assemble into Atg8-positive NE-localized puncta where NE protrusions containing nucleoplasmic material form. While many of these protrusions are eventually released into the cytosol, presumably as cargo enclosed in autophagosomes, some do not undergo this release. These abortive nucleophagy events may be caused by the presence of chromatin within the protrusions, as artificially tethering chromatin to

the INM strongly inhibits nucleophagy by preventing the release of NE protrusions (Fig. 7f).

All currently known autophagy receptors involved in nucleophagy exhibit relatively narrow species distributions. Atg39 is exclusively found in budding yeast species within the family *Saccharomycetaceae*, but not in the sister family *Saccharomycodaceae*, indicating that it originated no earlier than 152 million years ago[44]. Similarly, Epr1 is confined to the fission yeast genus *Schizosaccharomyces*[18], whose origin is dated around 207 million years ago[44]. Even more restricted is Npr1, which is found only in *S. pombe* and is absent in other fission yeast species (annotated by PomBase as an *S. pombe*-specific protein)[45], suggesting that its emergence occurred no earlier than 108 million years ago[44]. This frequent emergence of nucleophagy receptors during evolution may be explained by their minimal functional requirements—specifically, the presence of an AIM and localization to the ONM—as evidenced by the successful functional replacement of Epr1 and Npr1 with artificial fusion proteins that localize to the ONM and contain a cytosol-facing AIM.

The co-existence of Epr1 and Npr1 in *S. pombe*, despite their redundant roles in nucleophagy during nitrogen starvation, may be explained by their non-overlapping functions in other cellular contexts. Notably, Epr1 is essential for ER stress-induced ER-phagy and nucleophagy[18,19]. It is conceivable that Npr1 plays a critical role in nucleophagy under specific circumstances that were not examined in this study, highlighting the need for further investigation into its functions.

In *S. cerevisiae*, Atg39's role in nucleophagy extends beyond merely acting as a nucleophagy receptor. Through its single trans-membrane helix, which spans the ONM, and its C-terminal amphipathic helices, which are located in the NE lumen and bind to the INM, Atg39 establishes a physical connection between the ONM and the INM, facilitating NE deformation during nucleophagy[13,16]. In contrast, Npr1 and Epr1 appear to function solely as autophagy receptors, as their roles in nucleophagy can be substituted by either of two ONM-localized membrane proteins (Kms1 and Erg11) artificially fused with an AIM. In *S. pombe*, the NE-deforming function attributed to Atg39 in *S. cerevisiae* is likely carried out by other, yet-to-be-identified protein(s) involved in nucleophagy.

Previous studies have established that Atg39 is dispensable for the autophagic degradation of nuclear pore components in *S. cerevisiae*, with Nup159 instead acting as an autophagy receptor to facilitate this process[14–16]. Consistent with this, our study reveals that the autophagic degradation of nuclear pore components in *S. pombe* is independent of Epr1 and Npr1. Notably, our TurboID-Atg8 proximity labeling experiments showed enrichment of Nup146 (Supplementary Data 1), the *S. pombe* ortholog of *S. cerevisiae* Nup159. Further studies are needed to determine whether Nup146 serves as an autophagy receptor in *S. pombe*.

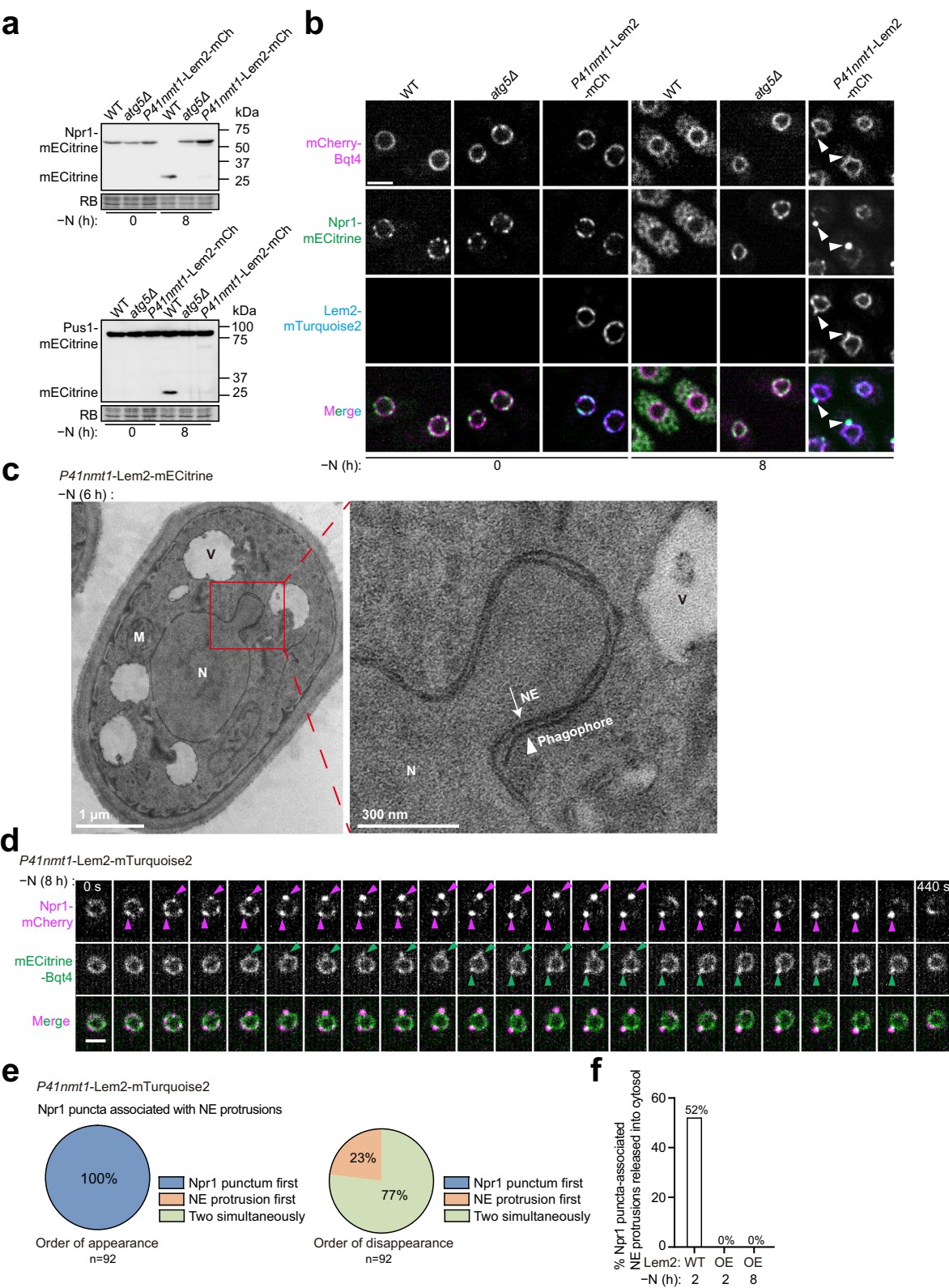

In Atg39-mediated nucleophagy in *S. cerevisiae*, two different models have been proposed for how NE fission gives rise to nucleus-derived vesicles that are enveloped within autophagosomes. One model posits that the INM and ONM protrude together towards the cytosol at the site where Atg39 assembles, subsequently undergoing simultaneous fission[13]. The alternative model suggests a two-step NE fission process: first, the INM undergoes fission, leading to the formation of INM-derived vesicles within the NE lumen; second, the ONM undergoes fission, a process that is partially dependent on the fission factor Dnm1[16,31]. Our EM analysis revealed that in *S. pombe*, the INM and ONM protrude together, suggesting that the simultaneous fission model may be applicable to *S. pombe*. The fission of the ONM may involve Yep1, the ortholog of human REEP1–4 proteins and *S. cerevisiae* Atg40[19]. However, the identity of

**Fig. 6 | Lem2 overexpression inhibits nucleophagy. a** Overexpression of Lem2-mCherry (mCh) from the exogenous *P41nmt1* promoter inhibited nucleophagy. Autophagic processing of Npr1-mECitrine and Pus1-mECitrine was analyzed by immunoblotting in wild-type (WT), *atg5Δ*, and Lem2-overexpressing (OE) cells. The experiment was independently repeated three times with similar results. **b** During nitrogen starvation, Npr1 puncta associated with NE protrusions (arrowheads) formed in Lem2-OE cells. Bar, 3 μm. The experiment was independently repeated three times with similar results. **c** A representative EM image of Lem2-OE cells after 6 h of nitrogen starvation shows a phagophore (arrowhead) wrapping around an NE protrusion. N nucleus, V vacuole, M mitochondrion. The experiment was independently repeated twice with similar results. **d** Time-lapse fluorescence

microscopy showed that Npr1 puncta (magenta arrowheads) in Lem2-OE cells exhibited kinetics of appearance and disappearance similar to those in wild-type cells. mECitrine-Bqt4 served as an NE marker, and NE protrusions are indicated by green arrowheads. Fluorescence images were captured at 20-s intervals. Bar, 2 μm. **e** The order of appearance and disappearance of Npr1 puncta and associated NE protrusions was analyzed in Lem2-OE cells after 8 h of nitrogen starvation. A total of 92 Npr1 puncta associated with NE protrusions were examined. **f** Quantification of the percentages of Npr1 puncta-associated NE protrusions released into the cytosol during time-lapse analysis. A total of 56 Npr1 puncta-associated NE protrusions in WT cells after 2 h of nitrogen starvation, 65 in Lem2-OE cells after 2 h of nitrogen starvation, and 92 in Lem2-OE cells after 8 h of nitrogen starvation were analyzed.

the INM fission factor(s) and the mechanism coupling ONM and INM fission remain unclear.

In *S. cerevisiae*, nucleophagy defects resulting from *atg39* deletion impair cell viability during nitrogen starvation[12]. This phenotype has been attributed to compromised Atg39-mediated degradation of Nvj1, a key micronucleophagy factor whose accumulation triggers excessive micronucleophagy[46]. In this study, we found that in *S. pombe*, nucleophagy defects caused by the deletion of *epr1* and *npr1* similarly result in decreased cell viability under nitrogen starvation. However, since *S. pombe* lacks Nvj1 and micronucleophagy has not been observed in this organism, the mechanisms driving viability loss are likely distinct. Notably, in *epr1Δ npr1Δ* mutant cells, we observed aberrant nuclear morphology characterized by membrane structures extending from the NE surface. This nuclear deformation allows for NE expansion without altering nuclear volume. The cause of this phenotype and the relationship between this phenotype and cell viability loss remain unclear. We speculate that an imbalance in the supply and turnover of NE components may underlie the observed NE extensions.

A central and unresolved question in the field of nucleophagy is how cells selectively degrade nuclear components without compromising genomic integrity. Our finding that the presence of chromatin in NE protrusions leads to abortive nucleophagy suggests a mechanism by which cells may safeguard genomic integrity during the degradation of nuclear components. Although this observation highlights a protective mechanism, the precise molecular basis for chromatin-mediated termination of nucleophagy remains to be elucidated. The failure of protrusions to release into the cytosol suggests that NE fission at the protrusion neck is inhibited. This inhibition could arise from two potential mechanisms: chromatin may passively obstruct membrane remodeling at the fission site, or dedicated surveillance systems may actively detect chromatin within protrusions and suppress NE fission. Further studies are needed to distinguish between these passive and active regulatory models.

## Methods
### Strain and plasmid construction
The *S. pombe* strains used in this study are listed in Supplementary Data 2, and the plasmids used in this study are listed in Supplementary Data 3. Unless stated otherwise, the strains were cultured in EMM medium at 30 °C. The compositions of the EMM medium, EMM−N medium, and other media are as described in ref. 47. Standard strain construction methods were used[47]. Deletion strains were generated through PCR-based gene targeting. The strains with Npr1, Epr1, or Lem2 endogenously tagged at their C-termini were constructed using PCR-based tagging[48].

Plasmids expressing proteins fused with TurboID and GFP-Atg8 under the *P41nmt1* promoter were constructed using modified pDUAL vectors[20,49]. The other protein-expressing plasmids were based on stable integration vectors (SIVs)[50,51]. These plasmids allow integration at specific loci, namely *ura4, ade6, lys3*, or *his5*. The plasmid expressing histone H3 (Hht2)-yeGFP was derived from the plasmid pDB5568[52], which was based on the pAde6^Pmel SIV plasmid and used a unique Pmel site as the linearization site. Other plasmids, designed to express

proteins fused with various N-terminal or C-terminal tags (mCherry, mScarlet2I, ymScarlet2I, mECitrine, or mTurquoise2), were based on modified SIV plasmids with a unique NotI site as the linearization site[51]. Plasmids expressing Npr1/Npr1(W22A/V25A)-mECitrine under its endogenous promoter were constructed by introducing the *npr1* promoter, together with the coding sequence of Npr1/Npr1(W22A/V25A), into a modified SIV containing the sequence encoding mECitrine. Plasmids expressing AIM^art-Npr1(30–244)-mCherry, AIM^art-Man1-mCherry, Rtn1-AIM^art-mCherry, AIM^art-mCherry-Kms1, AIM^art-Erg11-mCherry, and Erg11-AIM^art-mCherry were constructed by inserting the coding sequences of Npr1(30–244), Man1, Rtn1, Kms1, and Erg11, respectively, into a modified SIV containing the *P41nmt1* promoter and the sequence encoding AIM^art-mCherry. Plasmids expressing mCherry-Kms1-AIM^art were constructed by inserting the coding sequence of Kms1-AIM^art into a modified SIV containing the *P41nmt1* promoter and the sequence encoding mCherry. AIM^art corresponds to 3xEEEWEEL[29,30]. The plasmid expressing Lem2(LEM + BBM)-mECitrine was constructed by inserting the coding sequences of amino acids 1–60 and amino acids 261–279 of Lem2 into a modified SIV containing the *P41nmt1* promoter and the sequence encoding mECitrine[35,36]. Plasmids used to investigate the roles of LEM and BBM, including Lem2(LEM)-mCherry-Bqt4(TM), NLS-BD1-mCherry-Bqt4(TM), NLS-BD1*-mCherry-Bqt4(TM), NLS-Sso7d-mCherry-Bqt4(TM), NLS-Sso7d*-mCherry-Bqt4(TM), NLS-BD1-mCherry-h2NLS-LR2-WALP23, NLS-BD1*-mCherry-h2NLS-LR2-WALP23, NLS-Sso7d-mCherry-h2NLS-LR2-WALP23, and NLS-Sso7d*-mCherry-h2NLS-LR2-WALP23, were constructed by inserting the coding sequences of Bqt4(412–432) and h2NLS-LR2-WALP23 into modified SIVs, each containing the *P41nmt1* promoter and the sequences encoding Lem2(1–60), NLS-BD1, NLS-BD1(Y123F), NLS-Sso7d, or NLS-Sso7d(W24A/R43E), and mCherry[37–43]. The nucleotide sequence encoding NLS is ATGCCTAAGAAGAAGCGTAAGGTC. Plasmids that encode NLS-Sso7d-mCherry-Bqt3 under the endogenous promoter of Bqt3 were generated by inserting the *bqt3* promoter and the coding sequence of Bqt3 into a modified SIV containing the NLS-Sso7d and mCherry sequences. Plasmids expressing NLS-Sso7d-mCherry-Bqt3 under exogenous promoters were constructed by inserting the *P81nmt1* promoter, *P41nmt1* promoter, or *Pcyc1* promoter, along with the coding sequence of Bqt3, into a modified SIV containing the NLS-Sso7d and mCherry sequences.

Modified SIVs were used in this study to express proteins fused with GFP_{1-10} and 7×GFP_{11}[23–25]. The plasmid expressing Sum3-GFP_{1-10}-mCherry under the *Padh1* promoter was constructed by inserting the coding sequences of Sum3 and GFP_{1-10} into a modified SIV containing the *Padh1* promoter and mCherry sequences. The plasmid expressing Gbs1-GFP_{1-10}-mCherry under the *Padh1* promoter was constructed by inserting the sequences of GFP_{1-10} and mCherry between codons 496 and 497 of Gbs1 and placing the coding sequence of the fusion protein downstream of the *Padh1* promoter in an SIV plasmid. Plasmids expressing 7×GFP_{11}-Erg11, Erg11-7×GFP_{11}, 7×GFP_{11}-Npr1, and Npr1-7×GFP_{11} under the *P41nmt1* promoter were constructed by inserting the coding sequences of Erg11 or Npr1, together with the sequence encoding 7×GFP_{11}, into a modified SIV containing the *P41nmt1* promoter.

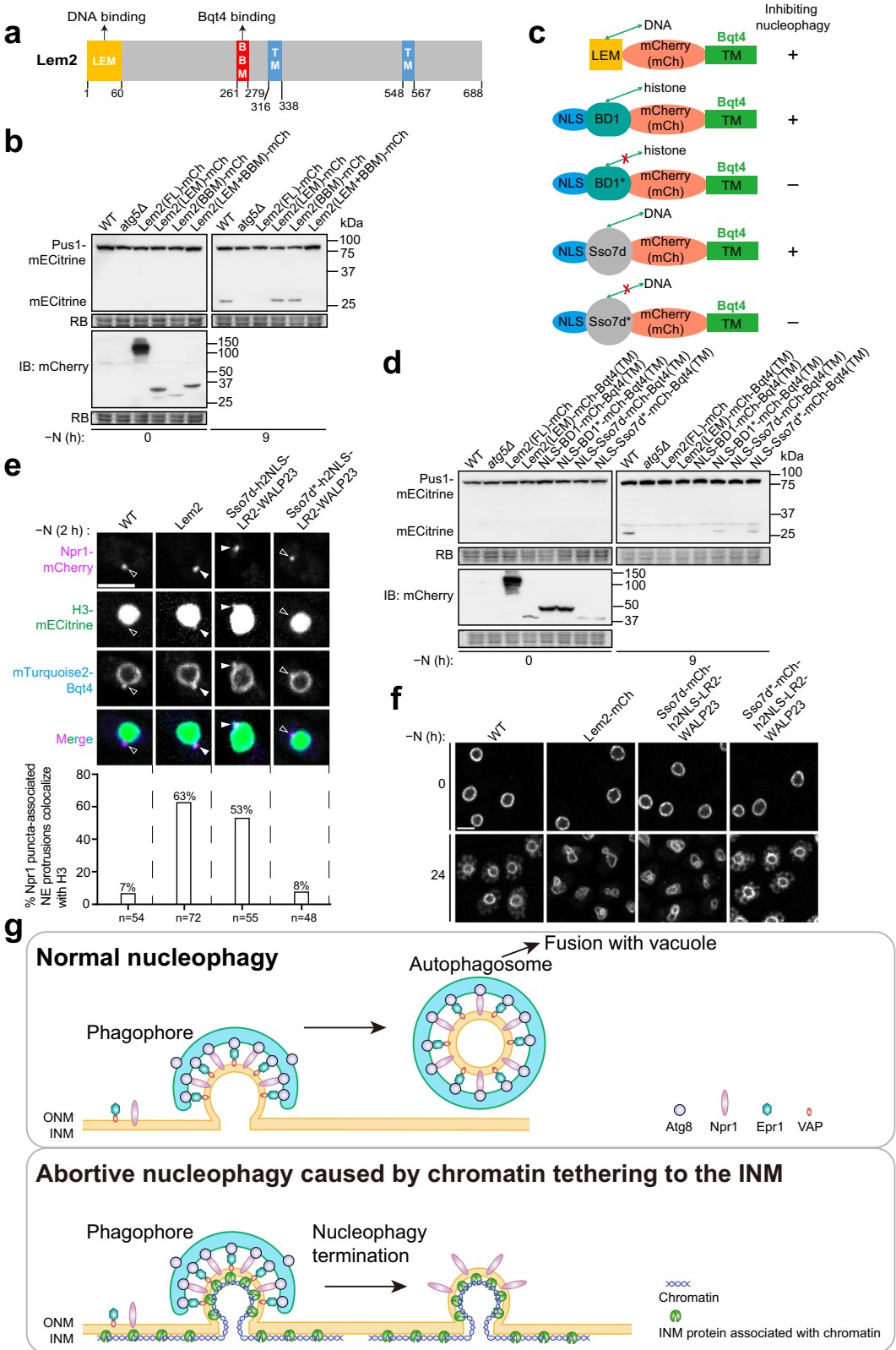

**g**

**Normal nucleophagy**

Phagophore

Autophagosome → Fusion with vacuole

ONM
INM

Atg8   Npr1   Epr1   VAP

**Abortive nucleophagy caused by chromatin tethering to the INM**

Phagophore

Nucleophagy termination

ONM
INM

Chromatin

INM protein associated with chromatin

Plasmids and strains generated in this study are available and can be requested from the corresponding author.

### TurboID-based proximity labeling and mass spectrometry analysis

Approximately 1000 $OD_{600}$ units of TurboID-mCherry (control) and TurboID-mCherry-Atg8 cells were collected following 4 h of nitrogen starvation. The collected cells were washed three times with deionized water and subsequently centrifuged to remove the supernatant. The resulting pellet was then reconstituted in 20 mL of deionized water, followed by the addition of 20 mL of 0.7 M sodium hydroxide. The mixture was incubated at room temperature on a rolling wheel for 10 min. The alkaline-treated cells were centrifuged to remove the supernatant. Afterward, 1 mL of lysis buffer (2% SDS, 0.06 M Tris-HCl,

**Fig. 7 | Chromatin tethering to the INM inhibits nucleophagy. a** Schematic of the functional domains and transmembrane helices (TMs) of Lem2. Two functional domains, the LEM domain (LEM) and the Bqt4-binding motif (BBM), are highlighted. **b** A fusion of the LEM and BBM domains of Lem2, but not either domain alone, inhibits nucleophagy. FL full length, mCh mCherry. The experiment was independently repeated three times with similar results. **c** Schematic of fusion proteins containing a DNA- or histone-binding domain from Lem2, Bdf1, or Sso7d, fused to the C-terminal transmembrane helix (TM) of Bqt4 (amino acids 412–432). Each fusion protein also includes a nuclear localization signal (NLS) and mCherry. BD1 refers to the first histone-binding bromodomain of Bdf1 (amino acids 66-208), while BD1* carries a Y123F mutation that disrupts histone binding. Sso7d is a non-specific DNA-binding protein from *Archaea*, while Sso7d* carries W24A and R43E mutations that impair its DNA-binding ability. **d** Fusion proteins containing the DNA- or histone-binding domains of Lem2, Bdf1, or Sso7d fused to the TM of Bqt4 inhibit nucleophagy in a DNA/histone binding-dependent manner. The expression levels of these mCherry (mCh)-tagged fusion proteins were analyzed by immunoblotting using an anti-mCherry antibody. The experiment was independently repeated three times with similar results. **e** Histone H3-mECitrine frequently co-

localized with Npr1 puncta-associated NE protrusions in Lem2-OE cells and Sso7d-h2NLS-LR2-WALP23-expressing cells. In contrast, wild-type cells and Sso7d*-h2NLS-LR2-WALP23-expressing cells exhibited minimal co-localization. h2NLS-LR2-WALP23 is an artificial INM protein comprising the NLS from the budding yeast protein Heh2 (h2NLS), a random sequence linker (LR2), and an artificially designed transmembrane helix peptide WALP23. Top: representative images in which Npr1 puncta-associated NE protrusions with co-localized H3 signals are indicated by solid arrowheads, while those without co-localized H3 signals are indicated by hollow arrowheads. Bar, 3 μm. Bottom: quantification of the percentages of Npr1 puncta-associated NE protrusions co-localized with H3. **f** Overexpression of Lem2 or expression of Sso7d-h2NLS-LR2-WALP23 resulted in NE morphology abnormalities during nitrogen starvation, but expression of Sso7d*-h2NLS-LR2-WALP23 did not. Fluorescence microscopy was used to visualize cells expressing the INM protein mECitrine-Bqt4 before and after 24 h of nitrogen starvation. Bar, 3 μm. The experiment was independently repeated three times with similar results. **g** Schematic models illustrating normal nucleophagy (top) and abortive nucleophagy resulting from chromatin tethering to the INM (bottom).

5% glycerol, and 4% 2-mercaptoethanol, pH 6.8) was added to resuspend the pellet, which was then incubated at 42 °C for 20 min. The supernatant was collected after centrifugation at 16,246×*g* for 30 min. For the purification of biotinylated proteins, approximately 100 μL of Streptavidin Agarose Resin (Thermo Fisher Scientific, Cat#20359) was used. The streptavidin beads were pre-washed twice using washing buffer A (50 mM Tris-HCl, 150 mM NaCl, 1 mM EDTA, 1 mM EGTA, 1% Triton X-100, 0.4% SDS, 1% NP40, and 1× Roche protease inhibitor cocktail, pH 7.5). After the addition of the streptavidin beads, the supernatant was incubated at room temperature on a rolling wheel for 3 h to allow the biotinylated proteins to bind to the streptavidin beads.

After incubation, the streptavidin beads were pelleted and washed twice with 1 mL of washing buffer A for 5 min and then incubated with 1 mL of washing buffer B (50 mM Tris-HCl, 2% SDS, pH 7.5) for 10 min. Subsequently, the beads were incubated twice with 1 mL of washing buffer A for 5 min each. Following this, biotinylated proteins were eluted from the beads by incubating the beads twice with 200 μL of elution buffer (50 mM Tris-HCl, 2% SDS, 5 mM biotin, pH 8.0) at 60 °C for 20 min, using a ThermoMixer C (Eppendorf) set at 1,000 × rpm.

To precipitate the proteins, 100 μL of 100% trichloroacetic acid (TCA) was added to approximately 400 μL of eluate. The mixture was then incubated overnight at 4 °C. Subsequently, the precipitated proteins were centrifuged at 16,246×*g* at 4 °C for 30 min, and the pellet was washed with acetone three times. The pellet was then resuspended with 30 μL of dissolution buffer (8 M urea, 100 mM Tris-HCl, pH 8.5) and dissolved by sonication using a water bath sonicator at room temperature for 15 min. Next, the proteins were reduced using 5 mM tris(2-carboxyethyl)phosphine at room temperature for 20 min, followed by alkylation with 10 mM iodoacetamide at room temperature for 15 min. The sample was diluted by a factor of 4 and digested into peptide fragments using trypsin at 37 °C overnight. To terminate the trypsin digestion, formic acid was added to a final concentration of 5%.

After the completion of the digestion process, LC-MS/MS analysis was conducted using an Easy-nLC II HPLC instrument (Thermo Fisher Scientific), which was coupled to a Q Exactive Orbitrap mass spectrometer (Thermo Fisher Scientific). A total of 8 μL of peptides were loaded onto a pre-column (100 μm ID, 4 cm long, packed with C18 10 μm 120 Å resin from YMC Co., Ltd) and separated on an analytical column (75 μm ID, 10 cm long, packed with Luna C18 1.8 μm 100 Å resin from Welch Materials) using an acetonitrile gradient from 0% to 30% over a duration of 100 min. The flow rate during the separation was maintained at 250 nL/min. From each full scan (resolution 70,000), the top 15 most intense precursor ions were selected for higher-energy collisional dissociation tandem mass spectrometry (HCD MS2) analysis,

with a normalized collision energy of 27 and a dynamic exclusion time of 30 s. The fragment ions obtained from the tandem mass spectrometry were detected using the Orbitrap in normal scan mode. Charge state rejection was enabled, and unassigned charge states, as well as charge states 1, 7, 8, and >8 were rejected. Mass spectrometry data were analyzed using pFind software, with a peptide false discovery rate (FDR) cutoff of 1%[53]. The TurboID-mass spectrometry experiment was independently repeated twice, yielding similar results.

## Immunoprecipitation

For immunoprecipitation (IP), approximately 100 OD$_{600}$ units of log-phase cells were collected and subjected to three washes with water. The cell pellet was mixed with 100 μL of lysis buffer A (50 mM HEPES, 1 mM EDTA, 150 mM NaCl, 10% glycerol, 3 mM DTT, 3 mM PMSF, Roche 3×Protease inhibitor cocktail, pH 7.5) and 800 μL of glass beads (BioSpec) with a diameter of 0.5 mm for cell lysis. The cells were lysed using the FastPrep-24 instrument at a speed of 6.5 m/s for 20 s. This lysis process was repeated three times. Then, 300 μL of lysis buffer B (50 mM HEPES, 1 mM EDTA, 150 mM NaCl, 10% glycerol, 0.05% NP40, 1 mM DTT, 1 mM PMSF, Roche 1×Protease inhibitor cocktail, pH 7.5) was added to the cell lysate. After centrifugation, 20 μL of the supernatant was retained as input, while the remainder was added to pre-washed GFP-Trap agarose beads at 4 °C for 3 h. The beads had been pre-washed with lysis buffer B. After centrifugation, the agarose beads were washed twice with wash buffer (50 mM HEPES, 1 mM EDTA, 150 mM NaCl, 10% glycerol, 0.05% NP40, 1 mM DTT, pH 7.5) and twice with lysis buffer B. The proteins bound to the GFP-Trap agarose beads were eluted by incubating them in SDS loading buffer (60 mM Tris-HCl, 4% SDS, 4% 2-mercaptoethanol, 5% glycerol, 0.002% bromophenol blue, pH 6.8) at 42 °C for 20 min for subsequent immunoblotting analysis.

## Fluorescence microscopy

We used cells cultured in liquid medium (EMM or EMM − N) for microscopy analysis. Live-cell imaging was performed using a Dragonfly 201-40 high-speed spinning-disk confocal microscope (Andor Technology), equipped with a 100×/1.4 NA objective lens, a Sona sCMOS camera, and two filter sets for mCherry/YFP/CFP and mCherry/GFP, respectively. Super-resolution images were acquired using a High Intelligent and Sensitive Structured Illumination Microscope (HIS-SIM, CSR Biotech Co., Ltd, Guangzhou), equipped with a 100×/1.5 NA objective lens, an sCMOS Flash 4.0 V2 camera, and a filter set for mCherry/GFP[54]. The microscopy images obtained were analyzed using Fiji[55].

For time-lapse imaging, a suspension of nitrogen-starved cells cultured in liquid medium (EMM − N) was applied to an agar pad

placed on a microscope glass slide (7.5 cm long)[56]. Firstly, two double-sided tapes were placed in the middle third of a clean glass slide, about 1.5 cm apart. Then, approximately 50 μL of hot melted agar (in EMM −N) was placed onto the glass slide, and immediately a coverslip (2.2 cm long) was placed on top of the agar drop. After the agar pad solidified in 3-5 min, the coverslip was removed, and 1 μL of the cell suspension was placed onto the agar pad. Time-lapse imaging analysis was performed after placing a new coverslip on top of the agar pad and ensuring the coverslip adhered tightly to the glass slide through the double-sided tapes.

## Immunoblotting-based protein processing assay

A total of 5 $OD_{600}$ units of cells were collected for lysis. The cells were resuspended in 300 μL of 20% TCA, and 700 μL of glass beads with a diameter of 0.5 mm were added. The cells were then lysed using the FastPrep-24 instrument at a speed of 6.5 m/s for 20 s, repeating this process three times. The cell lysates were transferred into new centrifuge tubes through centrifugation. To adjust the TCA concentration to 10%, 300 μL of deionized water was added to the cell lysates. The mixture was vortexed and centrifuged at $865{\times}g$ for 30 min. The resulting pellet was resuspended in SDS loading buffer (60 mM Tris-HCl, 4% SDS, 4% 2-mercaptoethanol, 5% glycerol, 0.002% bromophenol blue, pH 6.8) and incubated at 42 °C for 20 min. Subsequently, centrifugation was performed at $16{,}246{\times}g$ for 5 min, and the resulting supernatant was separated by 10% SDS-PAGE and subjected to immunoblotting using specific antibodies. The antibodies used for immunoblotting were anti-GFP mouse monoclonal antibody (1:3000 dilution, Roche, Cat#11814460001) and anti-mCherry rabbit polyclonal antibody (1:3000 dilution, ThermoFisher, Cat#PA5-34974). Post-immunoblotting staining of the PVDF membrane using Reactive Brown 10 (RB) served as the loading control[57]. For the statistical analysis of the protein processing data, Welch's $t$-test was performed using Excel spreadsheets downloaded from http://www.biostathandbook.com/twosamplettest.html [58].

## EM

For conventional transmission electron microscopy (TEM) analysis, a total of 50 $OD_{600}$ units of cells were collected after either 2 or 6 h of nitrogen starvation. The cells were fixed with glutaraldehyde and $KMnO_4$[59]. Following fixation, the cells underwent 13 rounds of water washing to remove any brownish particles. They were then dehydrated by passing through a series of graded ethanol solutions. Ultimately, the dehydrated samples were embedded in Spurr's resin[59]. Thin sections of 90 nm were examined using an FEI Tecnai G2 Spirit electron microscope equipped with a Gatan 895 4k × 4k CCD camera.

For EM analysis utilizing the genetically encoded EM tag MTn, samples containing 20 $OD_{600}$ units of vegetative cells expressing Npr1-mECitrine-MTn were processed[18,26].

For FIB-SEM, a total of 50 $OD_{600}$ units of cells were collected after 24 h of nitrogen starvation. Cell samples were processed using the same method as that used for preparing conventional TEM samples. The resin-embedded samples were mounted on aluminum stubs. FIB-SEM datasets were acquired using a Zeiss Crossbeam 550 microscope equipped with ATLAS 3D software (ZEISS). During the milling process with the focused gallium-ion beam, a milling current of 700 pA at 30 kV was used from the gallium emitter. The resin-embedded samples were milled in 10 nm layers. Scanning EM images were captured using an SE2 detector set at 2 kV and 1 nA. The image resolution in the xy plane was 10 nm/pixel. The alignment of image stacks, visualization, 3D reconstructions, movie creation, and quantification of nuclear volume, as well as the surface area and volume of the NE, were carried out using the Dragonfly Pro software (version 2022.2). For the statistical analysis, Student's two-sample $t$-test was performed using Excel spreadsheets downloaded from http://www.biostathandbook.com/twosamplettest.html.

## Y2H assay

The Y2H analysis was conducted using the Matchmaker system 3 (Clontech) to express two fusion proteins, namely the bait and prey. Bait plasmids were constructed by inserting the coding sequence of Atg8 into the pGBKT7 vector. Prey plasmids were constructed by inserting the coding sequence of Npr1/Npr1(W22A/V25A) into the pGADT7 vector. The AH109 yeast strain was co-transformed with the bait and prey plasmids and then selected on double dropout medium (SD/−Leu/−Trp). The activation of the *HIS3* and *ADE2* reporter genes was evaluated using quadruple dropout medium (SD/−Leu/−Trp/−His/−Ade). Photographs were taken after incubating the transformants on double dropout medium and quadruple dropout medium at 30 °C for 3 to 4 days.

## Growth phenotype assay (spot assay)

The *S. pombe* strains for the spot assay were inoculated into EMM medium supplemented with histidine, leucine, and uracil. The cells were cultured at 30 °C until they reached the logarithmic phase. A portion of the cells was collected, while the remaining cells were subjected to nitrogen starvation for 3 days and 7 days, respectively. Subsequently, the harvested cells were subjected to a five-fold dilution and spotted onto YES plates. The plates were then incubated at 30 °C and photographed after 3 days.

## Prediction of transmembrane topology and protein complex structure

The transmembrane topology was predicted using the CCTOP web server (https://cctop.ttk.hu)[22]. The structure of the Atg8-Npr1 complex was predicted using AlphaFold2-Multimer (version 2.3.1) with default parameters[27,28]. The structure exhibiting the highest confidence score among the predicted outputs was chosen for subsequent analysis. The visual representation of the predicted structure was generated using the Mol* Viewer (version 4.2.0)[60].

## Statistics and reproducibility

No statistical method was used to predetermine sample size. No data were excluded. The experiments were not randomized. The investigators were not blinded to allocation during experiments and outcome assessment. All experiments were independently repeated at least twice with similar results, indicating reproducibility. For the statistical analysis of the protein processing data, Welch's $t$-test was performed. For the statistical analysis of the FIB-SEM data, Student's two-sample $t$-test was performed.

## Reporting summary

Further information on research design is available in the Nature Portfolio Reporting Summary linked to this article.

## Data availability

The authors declare that all data supporting the findings of this study are available within the paper and its supplementary information files. Source data are provided with this paper.

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

## Acknowledgements

We are grateful to Ying Liu and He-Xia Luo of the NIBS EM Facility for their assistance with EM analysis. We also thank Wen-Yi Huang, Yin-Hua Lin, and Ke Du of Guangzhou Computational Super-Resolution Biotech Co., Ltd. for their support in live-cell imaging with their super-resolution microscope (HIS-SIM). Additionally, we thank Yi-Feng Jiang, Zhen-Hua Zhang, and Hao Zhang of the ZEISS Microscopy Customer Center in Beijing, as well as Eric Ho of Dragonfly, for their help with FIB-SEM analysis. We extend our gratitude to Meng-Li Shi for her assistance in drawing the models shown in Fig. 7g. This work was supported by the National Key R&D Program of China (2024YFA0917400 to L.-L.D.) and other grants from the Ministry of Science and Technology of China, the Beijing municipal government, and Tsinghua University to M.-Q.D. and L.-L.D.

## Author contributions

Conceptualization: Z.-H.M. and L.-L.D. Methodology and investigation: Z.-H.M., Z.-Q.P., Z.-D.J., G.-C.S., Y.H., F.S., C.-X.Z., Y.-F.J., M.-Q.D., and L.-L.D. Writing—original draft: Z.-H.M. and L.-L.D. Writing—review and editing: Z.-H.M. and L.-L.D. Funding acquisition: M.-Q.D. and L.-L.D.

## Competing interests

The authors declare no competing interests.
