## [Transparent Peer Review file · Nature Communications]

Nucleophagy is promoted by two autophagy receptors and inhibited by chromatin-nuclear envelope tethering in fission yeast

Corresponding Author: Professor Li-Lin Du

Version 0:

Reviewer comments:

Reviewer #1

(Remarks to the Author)

In this study, the authors investigated the molecular mechanisms of nucleophagy, selective autophagic degradation of nuclear components, in *Schizosaccharomyces pombe*. They identified a novel nucleophagy receptor, Npr1, an outer nuclear membrane protein that binds Atg8 and functions redundantly with Epr1, which was previously identified by the same group as a receptor for both the ER and the nucleus.

A particularly significant finding of this study is the discovery that chromatin tethering to the inner nuclear membrane (INM) inhibits nucleophagy. Based on the result that artificial chromatin-INM tethering impaired the release of nucleus-derived vesicles from NE protrusions, the authors propose that chromatin-INM tethering may serve as a safeguard mechanism that preserves genome integrity during nucleophagy. The authors also demonstrated that the *epr1-delta npr1-delta* double mutant, which is completely defective in nucleophagy, exhibits pronounced defects in nuclear morphology and reduced cell viability under starvation conditions, emphasizing the physiological importance of nucleophagy in this organism.

Thus, this work provides new insights into the mechanisms underlying nucleophagy. Overall, the study is well-designed and the data are compelling; however, the following issues should be addressed prior to publication.

1. On pages 12 and 13, the authors state that Atg8 “disappears” from NE protrusions and that Atg8 is deconjugated from the isolation membrane prior to its closure. However, this explanation appears to contradict their model shown in Fig. 7f, in which Atg8 is retained within autophagosomes. It seems that the Atg8 signal just decreases under levels undetectable by fluorescence microscopy at the late stages of nucleophagosome formation, so the authors should reconsider their wording (e.g., replacing “disappears” with “decreases”).

2. In Fig. 3a, degradation of nuclear proteins appears to be partially defective in *epr1-delta* cells and *npr1-delta* cells. To more precisely evaluate the effects of single deletion of *npr1* and *epr1*, the authors should quantify the results and perform statistical analysis.

Reviewer #2

(Remarks to the Author)

Reviewer #3

(Remarks to the Author)

The manuscript by Zhu-Hui Ma investigates the process of nucleophagy in fission yeast and reports that nucleophagy is promoted by two autophagy receptors and inhibited by chromatin-nuclear envelope (NE) tethering. The authors identify

Npr1 as an outer nuclear membrane (ONM) transmembrane protein that functions as a nucleophagy receptor by interacting with Atg8 via its N-terminal AIM. Npr1 functions redundantly with Epr1, and both receptors independently bind Atg8 on the ONM, consistent with genetic data showing that either receptor is sufficient to mediate nucleophagy.

The formation of Npr1 and Epr1 puncta with Atg8 at the NE appears to precede the development of NE protrusions, suggesting this may represent an early—possibly initiating—step in the process. While Atg8 eventually dissociates, Npr1 and Epr1 are captured by the autophagosome and delivered into vacuoles.

Npr1- and Epr1-mediated nucleophagy is required to maintain normal nuclear morphology, which, in turn, appears to contribute to cell survival during starvation. This part of the manuscript is compelling and the results are interesting and important. I would recommend concluding the manuscript at this point.

The authors also present evidence that chromatin tethering to the inner nuclear membrane interferes with or prevents the completion of nucleophagy. This second part of the study, which explores this chromatin tethering, is disconnected from the core findings. I suggest to either link it to mechanistically to Npr1, Epr1, or to remove this section. Currently the results in Figures 6 and 7 neither enhance nor diminishes the impact of the primary discoveries.

Major Points:

1. Genetic data clearly support redundant functions of Npr1 and Epr1 in nucleophagy. Mechanistically, however, this redundancy is less well explained and somewhat puzzling. Does loss of Npr1 result in upregulation of Epr1 (or vice versa)? What could be other compensatory mechanism?
2. The electron microscopy (EM) images of NE protrusions are compelling, but it would strengthen the manuscript if the authors could demonstrate the presence of Atg8, Npr1, or Epr1 on these structures.
3. The morphology of the NE protrusions observed by FIB-SEM should be described more clearly. In the images vacuolar structures appear to be captured by the NE protrusions. What is the role of vacuoles in the context of these protrusions?
4. In my view, the experiments involving overexpression of Lem2 are not essential, as they are not directly linked to Npr1 or Epr1. I recommend removing Figures 6 and 7 unless a mechanistic connection to Npr1 and Epr1 can be established.

Reviewer #4

(Remarks to the Author)

The work of Ma et al investigated mechanisms of nucleophagy in fission yeast. They identified a novel protein, Npr1, from Atg8 affinity purification, and established that it functions as a receptor for nucleophagy under starvation. They showed that Npr1 was present on the outer nuclear membrane, away from nuclear pores, with both termini facing the cytosol. They confirmed that Npr1 interacted with Atg8 via an AIM motif, and demonstrated via an elegant set of experiments that the main role of Npr1 is to bridge Atg8 with the nuclear envelope.

In live cell imaging, the authors found that Npr1 formed concentrated dots at a pace that was similar to Atg8. Their imaging data suggest that the concentrated Npr1 dots were pieces of nuclear envelope that eventually separated from the rest of nuclear envelope, possibly at the same time as they were sequestered into autophagosomes. The related timing, and the dependence of Npr1 dot formation on its interaction with Atg8, implies that the concentration of Npr1 occurs during the formation of autophagosomes. Under electron microscopy, a protruding piece of nuclear envelope could be seen wrapped with another piece of membrane, most likely autophagosome membrane.

In cells lacking nucleophagy receptors, nuclear envelope displayed irregular shapes, instead of spheres as in wild type cells. These mutants were also unable to survive starvation.

They further showed that when chromatin was artificially tethered to nuclear envelope, nucleophagy was blocked at a stage with protruding structures positive for Npr1, but unable to complete separation from the rest of nuclear envelope. Such a block could be achieved by over expressing a native protein with such tethering capacity, Lem2, or with artificially designed proteins.

Overall, the work of Ma et al reported a new autophagy receptor, and dissected how this protein functions. These novel discoveries will be of interest to autophagy and membrane biology field. However, there are several areas where the manuscript needs to be improved.

Main issues:

(1) How is nucleophagy triggered?

The authors' data analyzing the structural elements of Npr1 suggests that the sole function of a nucleophagy receptor is to link the substrate, the nuclear envelope, with the autophagosome membrane. In this case, nucleophagy is essentially a byproduct of general autophagy. But I am not convinced this is the whole story, and suggest the authors test the following: Observe Npr1 dependent nucleophagy under other autophagy inducing conditions. The above model predicts that in the presence of receptors, it will always occur whenever general autophagy occurs.

Target AIMS to other organelles will lead to their AIM dependent turnover under starvation. Try Golgi, peroxisomes, or even the plasma membrane?

Examine whether the interaction between Npr1 and Atg8 is constitutive or starvation induced.

(2) The physiological significance of nucleophagy.

The authors found that nuclear envelope displayed abnormal shapes in receptor mutants, and in atg5D mutant. To show that the phenotype is indeed due to nucleophagy, instead of some other functions of Npr1 or Atg8, the authors need to test if their artificial nuclear envelope – Atg8 tethers can recover the morphology and survival phenotypes, and other classes of atg mutants beyond the Atg8 system also have similar phenotypes.

The authors did not mention similar morphological abnormalities with AIM mutations, or with artificial chromatin tethering. Please confirm whether nuclear envelope morphology change is a shared consequence of all conditions blocking nucleophagy.

Besides shape, can the authors provide quantifications of the volumes and surface areas of nuclear envelopes with vs without nucleophagy? While detailed mechanistic investigation of nuclear envelope morphology change may be beyond the scope of this work, such simple quantifications may provide some clues as for the physiological function of nucleophagy.

Minor issues:

(1) The role of chromatin.

It is a very interesting observation that forcing chromatin into autophagosomes prevented their proper formation. However, the conditions the authors tested were all artificial. Is there a strong decrease in the degree of chromatin attachment to nuclear envelope from growing condition to starvation condition? In other words, what is the physiological relevance?

(2) The authors used several constructs when analyzing the membrane topology of Npr1. The functionality of these constructs need to be verified.

(3) In time lapse imaging, the authors observed about ¼ of the cases where no inner membrane protein dots were released into cytosol. The authors interpreted these events as premature termination of nucleophagy. My two alternative hypotheses are that the disappearance of Npr1 dots represents entry into vacuoles, and that the cases where no cytosolic inner membrane protein dots were observed represent cases with dots being dim to begin with. In particular for Npr1, their dots stayed at stable signal level before a complete disappearance at the next time point, unlike the gradual decline of Atg8, which implies Npr1 disappearance represents vacuole entry (similar to Ape1 in budding yeast). Can the authors provide more definitive evidence supporting their premature termination claim?

(4) In time lapse imaging, I do not see Npr1 signal on nuclear envelope, only dots. Is it because the nuclear envelope signal was too weak?

(5) When labeling lanes in immunoblots, the authors used a format that results in very long text lines. This makes it difficult to read and catch the most important information. The authors should change the labeling format.

Version 1:

Reviewer comments:

Reviewer #1

(Remarks to the Author)

The authors have addressed all the concerns we raised for the original manuscript in a satisfactory manner.

Reviewer #2

(Remarks to the Author)

Reviewer #3

(Remarks to the Author)

The authors have addressed most of my original concerns.

Reviewer #4

(Remarks to the Author)

The revised manuscript is substantially improved. The authors provided reasonable responses to most of my questions, and provided new data clarifying existing issues.

There is one comment that I feel the authors did not address properly. In my major point 1, I asked if nucleophagy was a byproduct of general autophagy given that the reported role of the receptor was solely to link the substrate membrane with Atg8, and one specific request was to test if the interaction between the receptor and Atg8 was induced by starvation. The authors responded by referring to a pre-existing CoIP data done under normal growing condition. While the data indeed demonstrates the existence of interaction under growing condition, it is fundamentally unrelated to whether the interaction is regulated by starvation or not.

Version 2:

Reviewer comments:

Reviewer #4

(Remarks to the Author)

The authors have addressed all my major concerns. I have no further comments.

REVIEWER COMMENTS

Reviewer #1 (Remarks to the Author):

In this study, the authors investigated the molecular mechanisms of nucleophagy, selective autophagic degradation of nuclear components, in *Schizosaccharomyces pombe*. They identified a novel nucleophagy receptor, Npr1, an outer nuclear membrane protein that binds Atg8 and functions redundantly with Epr1, which was previously identified by the same group as a receptor for both the ER and the nucleus.

A particularly significant finding of this study is the discovery that chromatin tethering to the inner nuclear membrane (INM) inhibits nucleophagy. Based on the result that artificial chromatin–INM tethering impaired the release of nucleus-derived vesicles from NE protrusions, the authors propose that chromatin–INM tethering may serve as a safeguard mechanism that preserves genome integrity during nucleophagy. The authors also demonstrated that the *epr1-delta npr1-delta* double mutant, which is completely defective in nucleophagy, exhibits pronounced defects in nuclear morphology and reduced cell viability under starvation conditions, emphasizing the physiological importance of nucleophagy in this organism.

Thus, this work provides new insights into the mechanisms underlying nucleophagy. Overall, the study is well-designed and the data are compelling; however, the following issues should be addressed prior to publication.

1. On pages 12 and 13, the authors state that Atg8 “disappears” from NE protrusions and that Atg8 is deconjugated from the isolation membrane prior to its closure. However, this explanation appears to contradict their model shown in Fig. 7f, in which Atg8 is retained within autophagosomes. It seems that the Atg8 signal just decreases under levels undetectable by fluorescence microscopy at the late stages of nucleophagosome formation, so the authors should reconsider their wording (e.g., replacing “disappears” with “decreases”).

Response: We thank the reviewer for the positive evaluation of our manuscript and for raising the concern regarding the interpretation of the “disappearance” of Atg8 puncta. We agree with the reviewer that not all Atg8 molecules are recycled; some must remain within the closed autophagosomes. We have added the following sentence to the main text of the revised manuscript: “It is important to note that the observed ‘disappearance’ of Atg8 does not imply that all Atg8 molecules are recycled, but rather that the level of Atg8 associated with the closed autophagosome falls below the detection limit of microscopy.”

2. In Fig. 3a, degradation of nuclear proteins appears to be partially defective in *epr1-delta* cells and *npr1-delta* cells. To more precisely evaluate the effects

of single deletion of *npr1* and *epr1*, the authors should quantify the results and perform statistical analysis.

Response: We have conducted the quantification and statistical analysis of the processing of Pus1-mECitrine, mECitrine-Bqt4, mECitrine-Nup82, and Ker1-mECitrine, as suggested by the reviewer. The results are presented in Supplementary Fig. 3a of the revised manuscript. As noted by the reviewer, the levels of processing were slightly reduced in the absence of *Epr1* or *Npr1*, but the *P* values are mostly greater than 0.05.

Reviewer #2 (Remarks to the Author):

Response: We thank the reviewer for contributing to the evaluation of our manuscript and for providing constructive suggestions.

Reviewer #3 (Remarks to the Author):

The manuscript by Zhu-Hui Ma investigates the process of nucleophagy in fission yeast and reports that nucleophagy is promoted by two autophagy receptors and inhibited by chromatin–nuclear envelope (NE) tethering. The authors identify *Npr1* as an outer nuclear membrane (ONM) transmembrane protein that functions as a nucleophagy receptor by interacting with *Atg8* via its N-terminal AIM. *Npr1* functions redundantly with *Epr1*, and both receptors independently bind *Atg8* on the ONM, consistent with genetic data showing that either receptor is sufficient to mediate nucleophagy.

The formation of *Npr1* and *Epr1* puncta with *Atg8* at the NE appears to precede the development of NE protrusions, suggesting this may represent an early—possibly initiating—step in the process. While *Atg8* eventually dissociates, *Npr1* and *Epr1* are captured by the autophagosome and delivered into vacuoles.

Npr1- and *Epr1*-mediated nucleophagy is required to maintain normal nuclear morphology, which, in turn, appears to contribute to cell survival during starvation. This part of the manuscript is compelling and the results are interesting and important. I would recommend concluding the manuscript at this point.

The authors also present evidence that chromatin tethering to the inner nuclear membrane interferes with or prevents the completion of nucleophagy. This second part of the study, which explores this chromatin tethering, is disconnected from the core findings. I suggest to either link it to mechanistically to Npr1, Epr1, or to remove this section. Currently the results in Figures 6 and 7 neither enhance nor diminishes the impact of the primary discoveries.

Response: We sincerely thank the reviewer for the thoughtful comments and for recognizing the significance of our findings on Npr1- and Epr1-mediated nucleophagy. We appreciate the suggestion to streamline the manuscript and agree that the chromatin tethering section of the manuscript (Figures 6 and 7) does not have a strong connection to the earlier parts.

However, we believe this section addresses a fundamental and unresolved question in the field: how do cells selectively degrade nuclear material via nucleophagy without risking genomic DNA loss? Our observation that artificial chromatin tethering to the inner nuclear membrane blocks nucleophagy provides the first experimental evidence that a mechanism may exist to safeguard against inadvertent chromatin degradation. To better integrate this section with the main narrative, we have revised the text in the Results and Discussion to emphasize chromatin-nuclear envelope association as a regulatory constraint on nucleophagy, rather than an independent phenomenon. We believe these revisions strengthen the conceptual link.

Major Points:

1. Genetic data clearly support redundant functions of Npr1 and Epr1 in nucleophagy. Mechanistically, however, this redundancy is less well explained and somewhat puzzling. Does loss of Npr1 result in upregulation of Epr1 (or vice versa)? What could be other compensatory mechanism?

Response: To address the reviewer's question, we have added Supplementary Fig. 3e in the revised manuscript, which shows that the loss of Npr1 did not affect the protein level of Epr1, and the loss of Epr1 did not affect the protein level of Npr1. We agree with the reviewer that the redundancy between Npr1 and Epr1 is intriguing and raises the question of why such redundancy arose during evolution. We speculate that under circumstances other than nitrogen starvation, these two proteins may not function redundantly and thus provide more opportunities for regulatory controls. Further investigations are needed to address this possibility.

2. The electron microscopy (EM) images of NE protrusions are compelling, but it would strengthen the manuscript if the authors could demonstrate the presence of Atg8, Npr1, or Epr1 on these structures.

Response: We thank the reviewer for the suggestion. Unfortunately, despite numerous attempts, we have not been able to identify EM sample preparation conditions that effectively preserve the structures of NE protrusions while also allowing for clear visualization of the MTn-tagged Atg8, Npr1, or Epr1. We hope to overcome these challenges in future experiments.

3. The morphology of the NE protrusions observed by FIB-SEM should be described more clearly. In the images vacuolar structures appear to be captured by the NE protrusions. What is the role of vacuoles in the context of these protrusions?

Response: We have revised the paragraph describing the morphology of the NE projections observed by FIB-SEM and have added a schematic diagram illustrating these structures in Fig. 5d of the revised manuscript. We also included the following sentence regarding their relationship with vacuoles: "Since an *S. pombe* cell contains numerous small vacuoles dispersed throughout the cytoplasm, the presence of vacuoles within the ring-like profiles of NE projections may simply result from this; however, we cannot rule out the possibility of a preferential spatial relationship between NE projections and vacuoles."

4. In my view, the experiments involving overexpression of Lem2 are not essential, as they are not directly linked to Npr1 or Epr1. I recommend removing Figures 6 and 7 unless a mechanistic connection to Npr1 and Epr1 can be established.

Response: We appreciate this kind suggestion. Although there is not a strong connection between the nucleophagy receptor section of the manuscript and the section on the inhibition of nucleophagy by chromatin-nuclear envelope tethering, we believe both sections contribute to a comprehensive understanding of nucleophagy. We have revised the text in the Results and Discussion to emphasize chromatin-nuclear envelope association as a regulatory constraint on nucleophagy, rather than an independent phenomenon. We believe these revisions strengthen the conceptual link.

Reviewer #4 (Remarks to the Author):

The work of Ma et al investigated mechanisms of nucleophagy in fission yeast. They identified a novel protein, Npr1, from Atg8 affinity purification, and established that it functions as a receptor for nucleophagy under starvation. They showed that Npr1 was present on the outer nuclear membrane, away from

nuclear pores, with both termini facing the cytosol. They confirmed that Npr1 interacted with Atg8 via an AIM motif, and demonstrated via an elegant set of experiments that the main role of Npr1 is to bridge Atg8 with the nuclear envelope.

In live cell imaging, the authors found that Npr1 formed concentrated dots at a pace that was similar to Atg8. Their imaging data suggest that the concentrated Npr1 dots were pieces of nuclear envelope that eventually separated from the rest of nuclear envelope, possibly at the same time as they were sequestered into autophagosomes. The related timing, and the dependence of Npr1 dot formation on its interaction with Atg8, implies that the concentration of Npr1 occurs during the formation of autophagosomes. Under electron microscopy, a protruding piece of nuclear envelope could be seen wrapped with another piece of membrane, most likely autophagosome membrane.

In cells lacking nucleophagy receptors, nuclear envelope displayed irregular shapes, instead of spheres as in wild type cells. These mutants were also unable to survive starvation.

They further showed that when chromatin was artificially tethered to nuclear envelope, nucleophagy was blocked at a stage with protruding structures positive for Npr1, but unable to complete separation from the rest of nuclear envelope. Such a block could be achieved by over expressing a native protein with such tethering capacity, Lem2, or with artificially designed proteins.

Overall, the work of Ma et al reported a new autophagy receptor, and dissected how this protein functions. These novel discoveries will be of interest to autophagy and membrane biology field. However, there are several areas where the manuscript needs to be improved.

Main issues:

(1) How is nucleophagy triggered?

The authors' data analyzing the structural elements of Npr1 suggests that the sole function of a nucleophagy receptor is to link the substrate, the nuclear envelope, with the autophagosome membrane. In this case, nucleophagy is essentially a byproduct of general autophagy. But I am not convinced this is the whole story, and suggest the authors test the following:

Observe Npr1 dependent nucleophagy under other autophagy inducing conditions. The above model predicts that in the presence of receptors, it will always occur whenever general autophagy occurs.

Target AIMS to other organelles will lead to their AIM dependent turnover under starvation. Try Golgi, peroxisomes, or even the plasma membrane?

Examine whether the interaction between Npr1 and Atg8 is constitutive or

starvation induced.

Response: We agree with the reviewer that nucleophagy is not simply a byproduct of general autophagy. Nucleophagy receptors are only one type of nucleophagy-specific factor that is important for nucleophagy but not general autophagy. Even though the nucleophagy role of the two fission yeast nucleophagy receptors, Npr1 and Epr1, can be largely substituted by an artificial tether connecting Atg8 to the nuclear envelope, other nucleophagy-specific factors are still indispensable. For example, we have previously shown that the fission yeast protein Yep1 (also known as Hva22 or Rop1), an ortholog of budding yeast Atg39 and human REEP1-4 proteins, is essential for nucleophagy but not for general autophagy (Zou et al. 2023, PMID:37939137). Budding yeast Atg39 appears to serve a combined role of fission yeast nucleophagy receptors and Yep1, thus fulfilling functions beyond merely linking Atg8 to the nuclear envelope. Given the complexity of the nucleophagy process, we anticipate that many additional nucleophagy-specific factors remain to be discovered.

Regarding autophagy-inducing conditions, we have identified only two that can robustly induce general autophagy in fission yeast: nitrogen starvation and DTT treatment, both of which also stimulate nucleophagy (Zou et al. 2023, PMID:37939137). It remains possible that there are undiscovered conditions that induce nucleophagy without inducing general autophagy in fission yeast.

The suggestion of artificially tethering Atg8 to other organelles is intriguing but lies beyond the scope of this study. We will pursue this line of investigation in the future.

The interaction between Npr1 and Atg8 is constitutive, as supported by co-immunoprecipitation analysis conducted prior to nitrogen starvation, as shown in Fig. 2b.

(2) The physiological significance of nucleophagy.

The authors found that nuclear envelope displayed abnormal shapes in receptor mutants, and in atg5D mutant. To show that the phenotype is indeed due to nucleophagy, instead of some other functions of Npr1 or Atg8, the authors need to test if their artificial nuclear envelope – Atg8 tethers can recover the morphology and survival phenotypes, and other classes of atg mutants beyond the Atg8 system also have similar phenotypes.

The authors did not mention similar morphological abnormalities with AIM mutations, or with artificial chromatin tethering. Please confirm whether nuclear envelope morphology change is a shared consequence of all conditions blocking nucleophagy.

Besides shape, can the authors provide quantifications of the volumes and

surface areas of nuclear envelopes with vs without nucleophagy? While detailed mechanistic investigation of nuclear envelope morphology change may be beyond the scope of this work, such simple quantifications may provide some clues as for the physiological function of nucleophagy.

Response: We thank the reviewer for suggesting these additional experiments to strengthen our conclusion. We have added new experimental data in Supplementary Fig. 5a-b showing that reintroducing either nuclear envelope-Atg8 tethers, AIM^{art}-Kms1 or Erg11-AIM^{art}, into the *epr1Δ npr1Δ* mutant largely restores NE morphology. Moreover, we have included data showing that either tether also restores the survival phenotype of the *epr1Δ npr1Δ* mutant (Supplementary Fig. 5d). AIM-mutated forms of Npr1 or Epr1 cannot rescue the NE morphology defect of the *epr1Δ npr1Δ* mutant (Fig. 5a and Supplementary Fig. 5a).

Additionally, we have added new data in Supplementary Fig. 5a, b showing that other autophagy mutants, including *atg1Δ*, *atg13Δ*, *atg9Δ*, *atg14Δ*, *atg2Δ*, *atg18aΔ*, and *atg18bΔ*, also exhibit NE morphology aberrations. In Fig. 7f and Supplementary Fig. 8, we show that either overexpression of Lem2 or expression of the artificial chromatin tether Sso7d-h2NLS-LR2-WALP23 leads to abnormal NE morphology.

Finally, we have added Supplementary Fig. 5c, which shows that the surface area and volume of the NE are increased in *epr1Δ npr1Δ* and *atg5Δ* cells compared to wild-type cells, whereas the volume of the nucleus remains largely unchanged. Thus, nuclear deformation allows for NE expansion without altering nuclear volume in nucleophagy-deficient cells.

Minor issues:

(1) The role of chromatin.

It is a very interesting observation that forcing chromatin into autophagosomes prevented their proper formation. However, the conditions the authors tested were all artificial. Is there a strong decrease in the degree of chromatin attachment to nuclear envelope from growing condition to starvation condition? In other words, what is the physiological relevance?

Response: In Discussion, we propose that the presence of chromatin in an NE protrusion being enclosed by an autophagic membrane results in abortive nucleophagy, and this serves as a safeguarding mechanism preventing the inadvertent degradation of genomic DNA during nucleophagy. Based on this model, we do not expect that the level of chromatin-NE association necessarily decreases under starvation conditions, because nucleophagy should be able to proceed as long as there are NE regions free of chromatin attachment. It would be interesting to determine whether the extent of chromatin-NE attachment changes when cells are transitioned from growth conditions to

starvation conditions. However, this cannot be easily accomplished, and therefore we chose not to pursue it during the revision.

(2) The authors used several constructs when analyzing the membrane topology of Npr1. The functionality of these constructs need to be verified.

Response: To address the reviewer's question, we added experimental data in Supplementary Fig. 3d examining whether various Npr1 fusion proteins retain the nucleophagy function. Reintroducing MTn-tagged Npr1 into the *epr1Δ npr1Δ* mutant fully restores the autophagic processing of Pus1-mCherry, whereas reintroducing Npr1 fused with GFP₁₁ at either the N-terminus or C-terminus into the *epr1Δ npr1Δ* mutant partially restores the autophagic processing of Pus1-mCherry. Thus, these fusions do not abrogate the function of Npr1.

(3) In time lapse imaging, the authors observed about ¼ of the cases where no inner membrane protein dots were released into cytosol. The authors interpreted these events as premature termination of nucleophagy. My two alternative hypotheses are that the disappearance of Npr1 dots represents entry into vacuoles, and that the cases where no cytosolic inner membrane protein dots were observed represent cases with dots being dim to begin with. In particular for Npr1, their dots stayed at stable signal level before a complete disappearance at the next time point, unlike the gradual decline of Atg8, which implies Npr1 disappearance represents vacuole entry (similar to Ape1 in budding yeast). Can the authors provide more definitive evidence supporting their premature termination claim?

Response: To address the reviewer's question, we imaged nitrogen-starved cells expressing Npr1-mCherry and the INM protein mECitrine-Bqt4 in a *fsc1Δ* background, where autophagosome-vacuole fusion is blocked. If the disappearance of Npr1 puncta indeed represents entry into vacuoles, we would expect to see a reduction in this type of Npr1 puncta and an increase in Npr1 puncta released into the cytosol in *fsc1Δ* cells compared to wild-type cells. What we observed is that in *fsc1Δ* cells, 50% of Npr1 puncta associated with NE protrusions exhibited simultaneous release into the cytosol, compared to 56% in wild-type cells. Additionally, 30% of puncta in *fsc1Δ* cells (27% in wild-type cells) disappeared simultaneously with the NE protrusion, while 20% in *fsc1Δ* cells (17% in wild-type cells) disappeared after the protrusion had vanished, with no subsequent detection of cytosolic puncta (Fig. 4i in the revised manuscript). These results suggest that the disappearance of NE protrusion-associated Npr1 puncta cannot be attributed to vacuole entry.

It is unclear why the disappearance of Npr1 appears abrupt rather than exhibiting a gradual decline in signal like Atg8. We speculate that this difference may be due to the distinct disassembly kinetics of the local assemblies where

these two proteins concentrate.

Furthermore, in our time-lapse imaging results, the signals of NE protrusions released into the cytosol are not initially stronger than those that vanished without ever being detected in the cytosol.

(4) In time lapse imaging, I do not see Npr1 signal on nuclear envelope, only dots. Is it because the nuclear envelope signal was too weak?

Response: The signal of Npr1 puncta in cells undergoing nucleophagy is much stronger than the signal of Npr1 on the nuclear envelope. In time-lapse imaging, to avoid bleaching, we used low laser power and short exposure times. Under such imaging conditions, the Npr1 puncta are clearly visible, but the signal of Npr1 on the nuclear envelope is no longer readily visible.

(5) When labeling lanes in immunoblots, the authors used a format that results in very long text lines. This makes it difficult to read and catch the most important information. The authors should change the labeling format.

Response: We thank the reviewer for the suggestion. We have made adjustments to enhance readability in our revised manuscript.

REVIEWER COMMENTS

Reviewer #1 (Remarks to the Author):

The authors have addressed all the concerns we raised for the original manuscript in a satisfactory manner.

Response: We thank the reviewer for the positive evaluation.

Reviewer #2 (Remarks to the Author):

Response: We thank the reviewer for the positive evaluation.

Reviewer #3 (Remarks to the Author):

The authors have addressed most of my original concerns.

Response: We thank the reviewer for the positive evaluation.

Reviewer #4 (Remarks to the Author):

The revised manuscript is substantially improved. The authors provided reasonable responses to most of my questions, and provided new data clarifying existing issues.

There is one comment that I feel the authors did not address properly. In my major point 1, I asked if nucleophagy was a byproduct of general autophagy given that the reported role of the receptor was solely to link the substrate membrane with Atg8, and one specific request was to test if the interaction between the receptor and Atg8 was induced by starvation. The authors responded by referring to a pre-existing CoIP data done under normal growing condition. While the data indeed demonstrates the existence of interaction under growing condition, it is fundamentally unrelated to whether the interaction is regulated by starvation or not.

Response: We thank the reviewer for highlighting this important point and acknowledge that our initial response did not adequately address whether the Npr1–Atg8 interaction is regulated by nitrogen starvation. To directly test this, we have now performed co-immunoprecipitation (CoIP) experiments comparing Npr1–Atg8 interaction levels under normal growth conditions and

after nitrogen starvation. As shown in the new Fig. 2b, Npr1 interacts with Atg8 to a comparable extent in both conditions, indicating that this interaction is constitutive and not induced or enhanced by starvation.